# An antagonism between Spinophilin and Syd-1 operates upstream of memory-promoting presynaptic long-term plasticity

Niraja Ramesh[1], Marc Escher[1], Oriane Turrel[1], Janine Lützkendorf[1], Tanja Matkovic[1], Fan Liu[2], Stephan J Sigrist[1]*

[1]Institute for Biology/Genetics, Freie Universität Berlin, Berlin, Germany; [2]Leibniz-Forschungsinstitut für Molekulare Pharmakologie, Berlin, Germany

**Abstract** We still face fundamental gaps in understanding how molecular plastic changes of synapses intersect with circuit operation to define behavioral states. Here, we show that an antagonism between two conserved regulatory proteins, Spinophilin (Spn) and Syd-1, controls presynaptic long-term plasticity and the maintenance of olfactory memories in *Drosophila*. While *Spn* mutants could not trigger nanoscopic active zone remodeling under homeostatic challenge and failed to stably potentiate neurotransmitter release, concomitant reduction of Syd-1 rescued all these deficits. The Spn/Syd-1 antagonism converged on active zone close F-actin, and genetic or acute pharmacological depolymerization of F-actin rescued the *Spn* deficits by allowing access to synaptic vesicle release sites. Within the intrinsic mushroom body neurons, the Spn/Syd-1 antagonism specifically controlled olfactory memory stabilization but not initial learning. Thus, this evolutionarily conserved protein complex controls behaviorally relevant presynaptic long-term plasticity, also observed in the mammalian brain but still enigmatic concerning its molecular mechanisms and behavioral relevance.

*For correspondence:
stephan.sigrist@fu-berlin.de

Competing interest: The authors declare that no competing interests exist.

## Editor's evaluation

The study advances mechanistic understanding of presynaptic plasticity (PHP): a process through which presynaptic nerve terminals adjust the strength of their output and which underpins key aspects of memory and learning. The authors use the well-characterised and tractable *Drosophila* NMJ to identify two key proteins – Spinophilin (Spn) and Syd-1 (a Rho GTPase activating protein) and show that the antagonism that occurs between these two components is sufficient to regulate F-actin stability at the synapse. Destabilization of F-actin, required for the maintenance of PHP, promotes synaptic vesicle release.

## Introduction

Synapses are key sites of information storage in the brain and shape computations of the nervous system. Their transmission strength is not hard-wired but plastically adapts to provide adequate input-output relationships, to maintain or restore transmission when compromised, and to store information (*Citri and Malenka, 2008*; *Takeuchi et al., 2014*; *Nicoll and Roche, 2013*; *Pozo and Goda, 2010*; *Marder and Goaillard, 2006*; *Turrigiano, 2012*; *Costa et al., 2017*). However, there is still a fundamental gap in our understanding of how dynamic changes of synapse performance intersect with circuit and sub-system activity to define behavioral states. This not the least is due to the inherent

complexity of mechanisms operating across different timescales (sub-second to lifetime), using a rich spectrum of pre- and postsynaptic homeostatic and Hebbian mechanisms.

In general, mechanisms of postsynaptic plasticity have been extensively worked out and often target the postsynaptic neurotransmitter receptors (*Nicoll and Roche, 2013*; *Herring and Nicoll, 2016*). Molecular cascades targeting the presynaptic SV release machinery to enhance transmission on a longer term ('sustained plasticity'), are much less characterized. There is broad evidence of presynaptic plasticity in the rodent brain, prominent at hippocampal mossy fiber bouton (MFB) synapses, involving a sequence of interleaved mechanistic components. Here, acute facilitation of release at presynaptic active zones (AZs) seemingly operates through changes in synaptic vesicle (SV) release probability and tighter $Ca^{2+}$ channel-sensor coupling, while longer-lasting changes required for stable memory formation at the MFB have been shown to occur via the addition of novel SV release sites (*Monday et al., 2018*; *Monday et al., 2022*), AZ $Ca^{2+}$ channel accumulation (*Fukaya et al., 2021*) and long-lasting readily releasable SV pool (RRP) size increases (*Vandael et al., 2020*). The upstream regulatory control mechanisms which bring about these changes remain unknown, however, hampering genetic access and inquiry into their behavioral relevance.

Concerning the elucidation of presynaptic plasticity mechanisms, *Drosophila* neuromuscular junction (NMJ) synapses are uniquely suited as they are accessible to a combination of genetics, pharmacology, electrophysiology and super-resolution imaging. Notably, robust presynaptic homeostatic plasticity (PHP) evident in an increase of SVs being released per action potential can be induced at NMJ synapses via a pharmacological blockade of the postsynaptic glutamate receptors (GluRs). In the multi-step course of the NMJ PHP process, increase in AZ scaffold proteins Bruchpilot (BRP, an ELKS/CAST homolog) (*Böhme et al., 2019*; *Goel et al., 2017*; *Weyhersmüller et al., 2011*), Rim-BP, munc13 family member Unc13A (*Böhme et al., 2019*), and voltage-gated $Ca^{2+}$ channel Cacophony (*Gratz et al., 2019*) provoke increases in $Ca^{2+}$ transients (*Müller and Davis, 2012*), and an increase in the size of the RRP (*Weyhersmüller et al., 2011*). Important for the work presented here, we recently found that BRP and Unc13A are upregulated during memory formation at *Drosophila* mushroom body (MB) lobe AZs, with this AZ remodeling being needed to stabilize new memories but being dispensable for initial learning (*Turrel et al., 2022*).

In this work, we show that Spinophilin (Spn), known to function during AZ development in promoting BRP accumulation (*Ramesh et al., 2021*), also executes AZ remodeling during PHP and is crucial for sustaining PHP over longer timescales. Syd-1, which antagonizes Spn function during development (*Ramesh et al., 2021*), also antagonizes Spn function in AZ remodeling and sustaining PHP. Spn functions by modulating the actin cytoskeleton (*Chia et al., 2012*; *Nakanishi et al., 1997*) which regulates access of SVs to release sites. Spn and Syd-1 seemingly also function antagonistically in the MB memory center in sustaining memories. Having identified this evolutionarily conserved regulatory complex as functioning high in the regulatory scheme of presynaptic long-term plasticity via actin remodeling might pave the way towards addressing the behavioral role of presynaptic long-term plasticity in the mammalian brain.

## Results

Homeostatic synaptic plasticity serves to maintain baseline transmission strength in response to altered pre- or postsynaptic function. It is conserved from invertebrates through humans, but perhaps best illustrated in *Drosophila* NMJ synapses (*Davis and Müller, 2015*). PHP at NMJ synapses can be triggered by the application of the GluR blocker Philanthotoxin (PhTx) resulting in a compensatory enhancement of presynaptic neurotransmitter release (*Davis and Müller, 2015*; *Frank, 2014*; *Lazarevic et al., 2013*). PhTx application initiates presynaptic AZ remodeling within minutes, observed as increased AZ cytomatrix proteins. Notably however, absence of BRP still allowed for functional potentiation in the rapid induction phase (10 min PhTx treatment; *Böhme et al., 2019*) but could not sustain potentiation during the maintenance phase (30 min PhTx treatment; *Turrel et al., 2022*) of PHP, or in combination with *glurIIA* mutant, a genetic model of long-term PHP (*Böhme et al., 2019*).

## Presynaptic Spinophilin functions to sustain homeostatic plasticity through active zone remodeling

In order to identify regulators specific for AZ remodeling and the maintenance phase of PHP, we analyzed mutants of the evolutionarily conserved AZ regulator, Spn (*Muhammad et al., 2015*), needed for efficient BRP incorporation during the developmental assembly of AZs (*Ramesh et al., 2021*). *Spn* mutants indeed failed to show any BRP increase when challenged with PhTx (*Figure 1a–b*) while control animals showed a robust increase. We first considered whether this inability might be a sheer consequence of its developmental phenotype. We therefore focused on mutants of Spn's binding partner Neurexin-1 (Nrx-1), as well as Nrx-1's transsynaptic binding partners Neuroligin-1 (Nlg1) and Nlg2, all of which have impaired developmental AZ assembly (*Ramesh et al., 2021*; *Sun et al., 2011*; *Banovic et al., 2010*; *Owald et al., 2012*; *Zeng et al., 2007*), with specific emphasis on *Nlg2* mutants which phenocopy *Spn* mutants in terms of developmental assembly (*Ramesh et al., 2021*). We evaluated BRP levels upon 10 min PhTx treatment at the respective mutant NMJs and found that *Nrx-1* (*Figure 1—figure supplement 1c–d*), *Nlg1* (*Figure 1—figure supplement 1e–f*) and importantly *Nlg2* (*Figure 1—figure supplement 1g–h*) mutants, all showed increased BRP incorporation similar to controls (*Figure 1—figure supplement 1a–b*), indicating that the developmental and PhTx-triggered BRP incorporation can be mechanistically separated (see discussion). Our subsequent analysis unmasked a regulatory role of Spn also concerning the incorporation of the release-crucial Unc13A at PhTx-challenged NMJs (*Figure 1c–d*).

Given Spn's function for incorporating BRP during PHP, and BRP's specific role in PHP maintenance (not PHP induction) (*Turrel et al., 2022*), we asked whether Spn would also specifically function in PHP maintenance (30 min PhTx treatment) and not induction (10 min PhTx treatment). Indeed, this is exactly what we found. PhTx treatment resulted in a reduction in mini amplitude at both time points (*Figure 1e–g and j*). During induction, both controls but also *Spn* mutants showed compensation in evoked release (*Figure 1e and h*) and the number of SVs released per action potential (quantal content, QC (*Figure 1e and i*)). In contrast, during the maintenance phase, while evoked release was compensated and QC increased in controls, *Spn* mutants failed to do so (*Figure 1f and k*). In fact, *Spn* mutants showed even a significant reduction in QC when challenged with PhTx for 30 min (*Figure 1f and l*).

Spn at the NMJ is expressed not only at presynaptic AZs of the motoneurons but also at the postsynaptic site (*Muhammad et al., 2015*). Given the presynaptic nature of its plasticity deficit, we asked whether presynaptic Spn was responsible for PHP. Thus, we conducted a rescue experiment and, in *Spn* mutants, re-expressed full-length Spn exclusively in motor neurons using the Ok6-Gal4 driver (*Figure 1—figure supplement 2*). Indeed, both the deficit of BRP incorporation (*Figure 1—figure supplement 2a–b*) as well as QC increase after 30 min PhTx challenge (*Figure 1—figure supplement 2c–i*) were both fully rescued, suggesting that presynaptic Spn is sufficient to mediate BRP increase and PHP maintenance. Full-length Spn, exclusively expressed in the postsynaptic muscle cells, using the Mef2-Gal4 driver, did not rescue the PHP-dependent BRP incorporation deficit of *Spn* mutants (data not shown). Moreover, in PhTx-untreated controls, *Spn* mutants with muscle expression of full-length Spn had reduced evoked amplitudes when compared to controls (*Figure 1—figure supplement 2n*). Upon PhTx treatment, *Spn* mutants with muscle expression of full-length Spn showed a modest increase in quantal content (*Figure 1—figure supplement 2c*, j-o), however, not to the same extent as with motor neuron expression of Spn. Thus, muscle expression of Spn also seems able to somewhat milden the complete loss of Spn for PHP, albeit through mechanisms independent of BRP. We therefore focused our analysis on presynaptic Spn in the following experiments.

## The Spinophilin/Syd-1 antagonism controls active zone structural and functional plasticity

We next considered that if Spn operated in a truly regulatory manner in AZ plasticity, it might be opposed by counteracting regulatory activities, and identifying these could enrich our mechanistic understanding of AZ remodeling and plasticity. In this context, we recently discovered that another AZ residing conserved protein, Syd-1, operates antagonistic to Spn in AZ development (*Ramesh et al., 2021*). We asked whether this relationship was co-opted in support of sustained PHP. BRP immunostaining after 10 min of PhTx-treatment showed that *Syd-1* mutants showed robust BRP increase (*Figure 2a–b*), with a trend towards overcompensation compared to controls (116.61% that

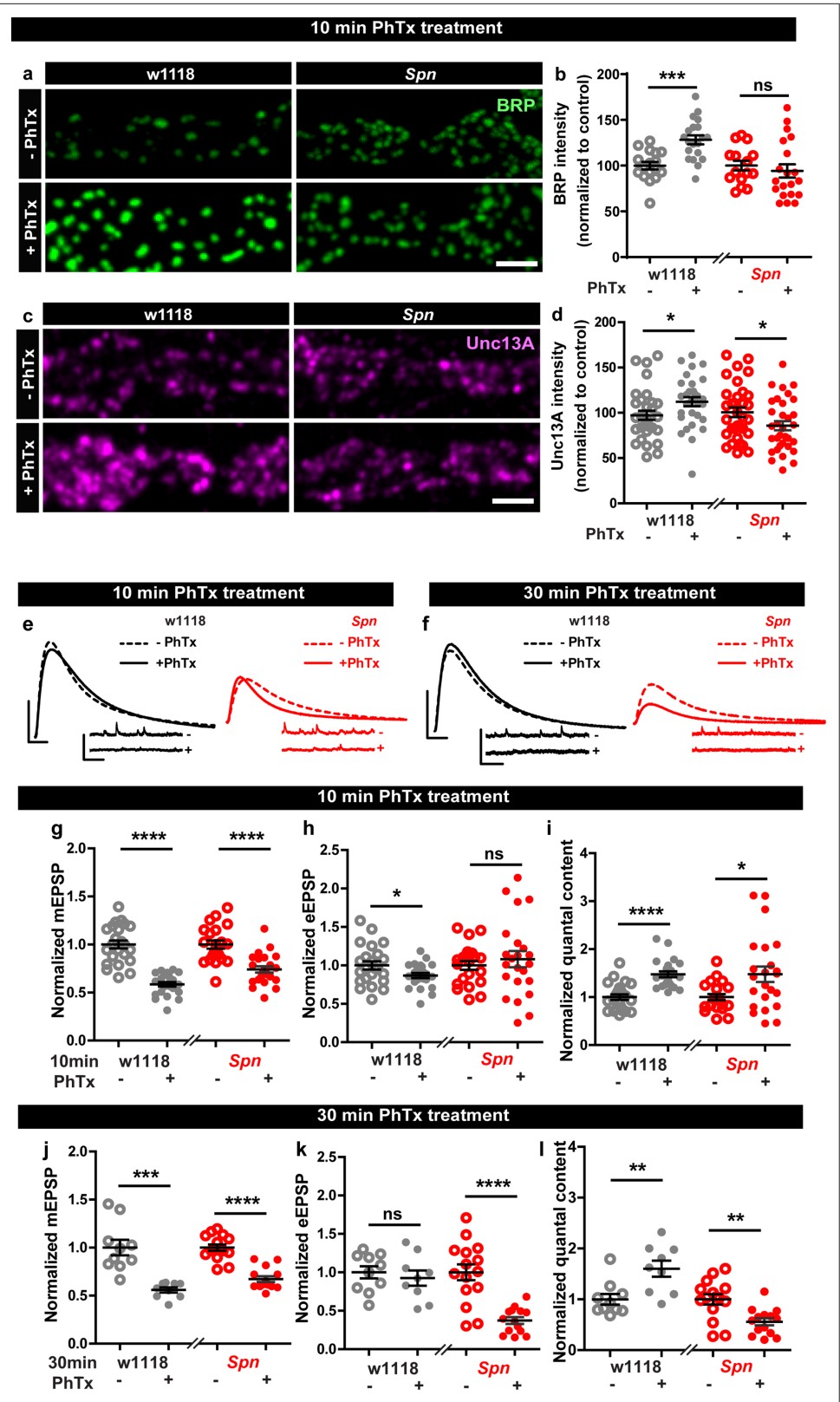

**Figure 1.** Presynaptic Spinophilin functions to sustain homeostatic plasticity through active zone remodeling. (**a,c**) Representative images of third-instar larval muscle 4 NMJs immunostained with antibodies against BRP and Unc13A. Scale bars: 2 μm. (**a,d**) Control animals show an increase in BRP and Unc13A levels upon PhTx treatment while Spn mutants do not (**b,d**) mean BRP and Unc13A intensities measured through confocal imaging.

*Figure 1 continued on next page*

*Figure 1 continued*

(**e,f**) Representative traces of eEJP and mEJP measurements at third-instar larval muscle 6/7 NMJs. Scale bars: eEJP, 10 ms, 10 mV; mEJP traces 500 ms, 5 mV. (**g,j**) mEJP amplitudes are reduced upon PhTx treatment. eEJP amplitudes are compensated (**h**) and QC is increased (**i**) in control and Spn mutants upon 10 min PhTx treatment. eEJP amplitudes are compensated (**k**) and QC is increased (**l**) in control but not Spn mutants upon 30 min PhTx treatment. Also see *Figure 1—figure supplements 1 and 2*. Source data as exact normalized and raw values, detailed statistics including sample sizes and P values are provided in *Figure 1—source data 1*. *p ≤ 0.05; **p ≤ 0.01; ***p ≤ 0.001; n.s., not significant, p > 0.05. All panels show mean ± s.e.m.

The online version of this article includes the following source data and figure supplement(s) for figure 1:

**Source data 1.** Table containing exact values for the data depicted in *Figure 1*, along with details of statistical analyses.

**Figure supplement 1.** *Nrx-1*, *Nlg1* and *Nlg2* mutants undergo homeostatic plasticity, while *Spn* mutants cannot sustain homeostatic plasticity.

**Figure supplement 1—source data 1.** Table containing exact values for the data depicted in *Figure 1—figure supplement 1*, along with details of statistical analyses.

**Figure supplement 2.** Presynaptic Spinophilin is sufficient to sustain homeostatic plasticity.

**Figure supplement 2—source data 1.** Table containing exact values for the data depicted in *Figure 1—figure supplement 2*, along with details of statistical analyses.

of controls). In line with the antagonistic roles of Spn and Syd-1, removing a single gene copy of *syd-1* from *Spn* mutants resulted in the reinstatement of BRP incorporation in *Spn* mutants (*Figure 2c–d*).

Upon PhTx challenge, *Syd-1* mutants showed a very robust QC increase at 30 min (*Figure 2e and i*) resulting from the normal PhTx-induced reduction in mini amplitudes (*Figure 2e and g*) and compensation of evoked release (*Figure 2e and h*). Indeed, we recognized a tendency towards an over-compensation of evoked response and QC compared to controls (129.26% that of controls). To work out this potential hyperplasticity phenotype of *Syd-1* mutants, we followed up on BRP levels upon 30 min PhTx challenge. Interestingly, within this time interval, BRP levels returned to pre-PhTx baseline levels in controls (*Figure 2m–n* grey). Thus, BRP levels seem to increase only transiently upon PhTx-induced AZ remodeling, similar to the transient BRP increases triggered by Pavlovian olfactory conditioning in the mushroom body (*Turrel et al., 2022*). Notably, different from controls, *Syd-1* mutants continued to show an increase in BRP levels at 30 min (*Figure 2m–n*). Similarly, presynaptic overexpression of Spn showed a trend towards higher BRP incorporation upon PhTx treatment than control terminals (*Figure 1—figure supplement 2a–b*).

Thus, a finely balanced counterplay between these two antagonistically operating presynaptic regulators, Spn and Syd-1, seems to set the limits of both structural AZ remodeling and functional plasticity here, with absence of Spn abrogating but absence of Syd-1 boosting the sustained component of AZ-mediated plasticity. This antagonism we identify, once again suggests a presynaptic action of Spn in this context, as Syd-1 expression and function have exclusively been reported at the presynapse (*Owald et al., 2012*; *Hallam et al., 2002*; *Wentzel et al., 2013*).

## The Spinophilin/Syd-1 antagonism converges on presynaptic F-actin dynamics

Since the absence of Spn could be offset by reducing Syd-1 levels, these proteins likely antagonistically affect a downstream signal. In order to identify possible downstream mechanisms through which Spn might mediate its function during PHP, we identified proteins enriched in BRP- and Spn-immunoprecipitations (IPs) from *Drosophila* brain synaptosomes (*Depner et al., 2014*) through mass spectrometry. As a validation of the list of proteins that were returned as interaction partners of Spn in this work, we were able to reconfirm previously known (*Muhammad et al., 2015*), for example, Syd-1 (*Figure 3b*) and Nrx-1 (not shown). We concentrated on proteins which were exclusively and significantly enriched in Spn (but not BRP) IPs, following the logic that interactions unique to Spn might be responsible for its distinctive functions at the AZ, rather than just reflect its AZ localization in the AZ and BRP complex. Comparative functional enrichment analysis of the Spn (*Figure 3a*) interactome suggested that Spn was specifically associated with translational control. To address whether translation would mediate PHP at the NMJ, we either fed larvae or treated them acutely with cycloheximide

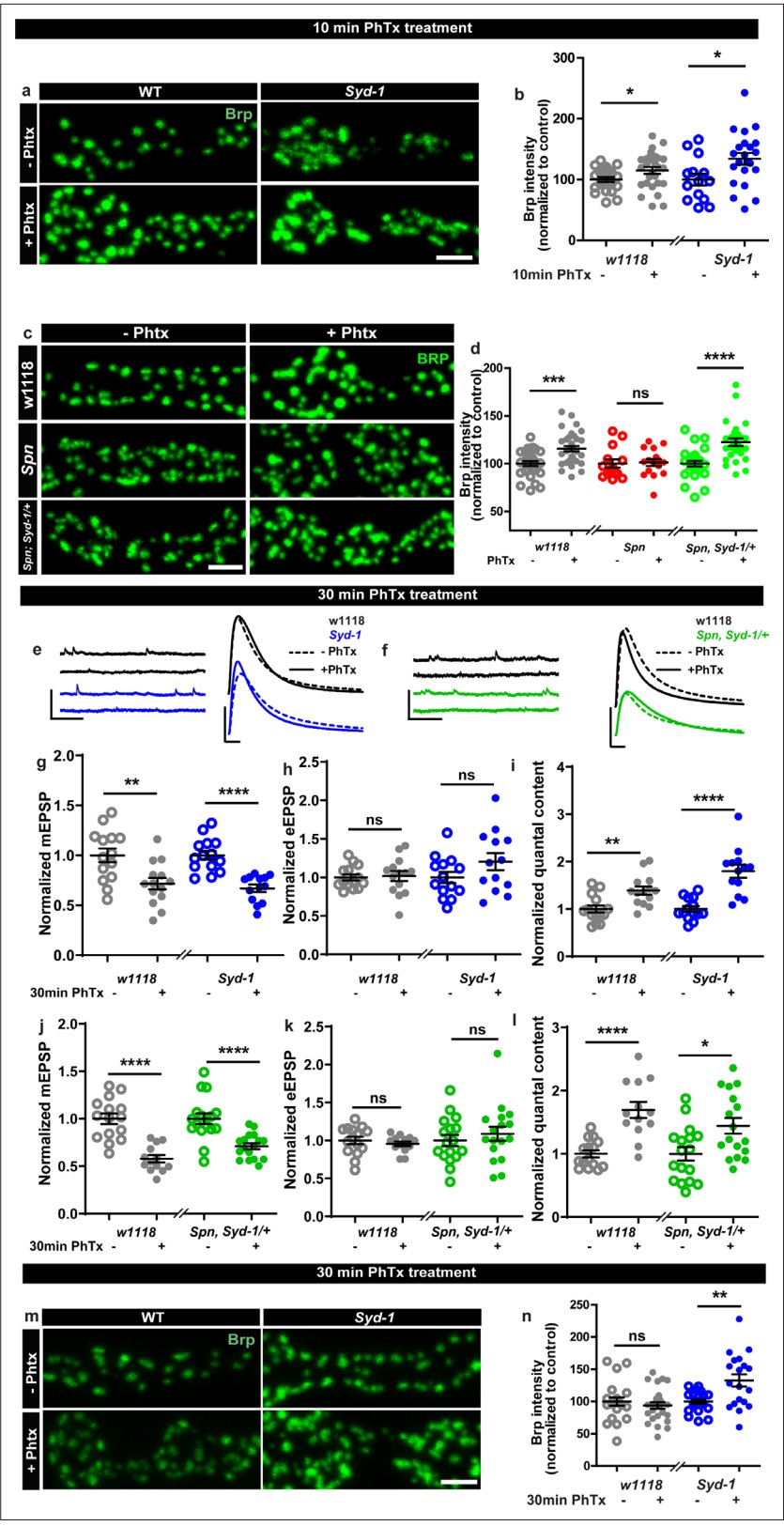

**Figure 2.** Spinophilin/Syd-1 antagonism controls active zone structural remodeling and functional plasticity. (**a,c,m**) Representative images of third-instar larval muscle 4 NMJs immunostained with an antibody against BRP. Scale bars: 2 μm. (**a–b**) Control and *Syd-1* mutant animals show an increase in BRP upon 10 min PhTx treatment (**c–d**) Removing a single *syd-1* gene copy from *Spn* mutants (*Spn,Syd-1/+*) reinstates the BRP increase upon 10

*Figure 2 continued on next page*

*Figure 2 continued*

min PhTx treatment (**b,d**) mean BRP intensities measured through confocal imaging. (**e,f**) Representative traces of mEJP and eEJP measurements at third-instar larval muscle 6/7 NMJs. Scale bars: eEJP, 10 ms, 10 mV; mEJP traces 500 ms, 5 mV. (**g,j**) mEJP amplitudes are reduced upon PhTx treatment. eEJP amplitudes are compensated (**h**) and QC is increased (**i**) in control and *Syd-1* mutants upon 30 min PhTx treatment. eEJP amplitudes are compensated (**k**) and QC is increased (**l**) in control and single copy s*yd-1* in *Spn* (*Spn,Syd-1/+*) mutants upon 30 min PhTx treatment. (**m–n**) *Syd-1* mutant animals continue to show an increase in BRP upon 30 min PhTx treatment, while controls do not. Also see *Figure 2—figure supplement 1*. Source data as exact normalized and raw values, detailed statistics including sample sizes and P values are provided in *Figure 2—source data 1*. *p ≤ 0.05; **p ≤ 0.01; ***p ≤ 0.001; n.s., not significant, p > 0.05. All panels show mean ± s.e.m.

The online version of this article includes the following source data and figure supplement(s) for figure 2:

**Source data 1.** Table containing exact values for the data depicted in *Figure 2*, along with details of statistical analyses.

**Figure supplement 1.** Spinophilin/Syd-1 antagonism sustains homeostatic plasticity.

**Figure supplement 1—source data 1.** Table containing exact values for the data depicted in *Figure 2—figure supplement 1*, along with details of statistical analyses.

---

(a translation elongation blocker) and then challenged them with PhTx. However, the animals did not show impaired PHP and could upregulate their QC (*Figure 3—figure supplement 1a–h*), suggesting that acute translation is not required to sustain PHP measured at 30 min of PhTx treatment. Given the lack of assays allowing for acute in vivo monitoring of local neuronal translation in our system, we cannot be fully sure about the effectiveness of our intervention, however.

Notably, Spn is known to interact with F-actin and contributes to F-actin remodeling during synapse development (*Chia et al., 2012*; *Nakanishi et al., 1997*; *Ryan et al., 2005*) and F-actin bundling in dendritic filopodia (*Satoh et al., 1998*). Indeed, our Spn IPs were enriched for various regulators of F-actin dynamics (labeled blue in *Figure 3b*). In contrast, BRP IPs were rather characterized by factors semantically associated with synaptic signaling and AZ organization (data not shown).

We thus went on to analyze the F-actin status at presynaptic AZs of *Spn* mutants. Keeping a potential antagonistic mechanism with Syd-1 in mind, we expressed GFP-labeled actin within larval motor neurons (Ok6-Gal4 driver) in both *Spn* and *Syd-1* mutants. *Spn* AZs accumulated atypically high actin levels (*Figure 4a–b*), while *Syd-1* mutants showed a reduction (*Figure 4c–d*) compared to controls. We next conducted intravital (imaging NMJs in intact living larvae through their cuticle) fluorescence recovery after photobleaching (FRAP) experiments (*Figure 4e*; *Andlauer and Sigrist, 2012*). These FRAP experiments indicated higher exchange of AZ actin over time in *Spn* mutants when compared to

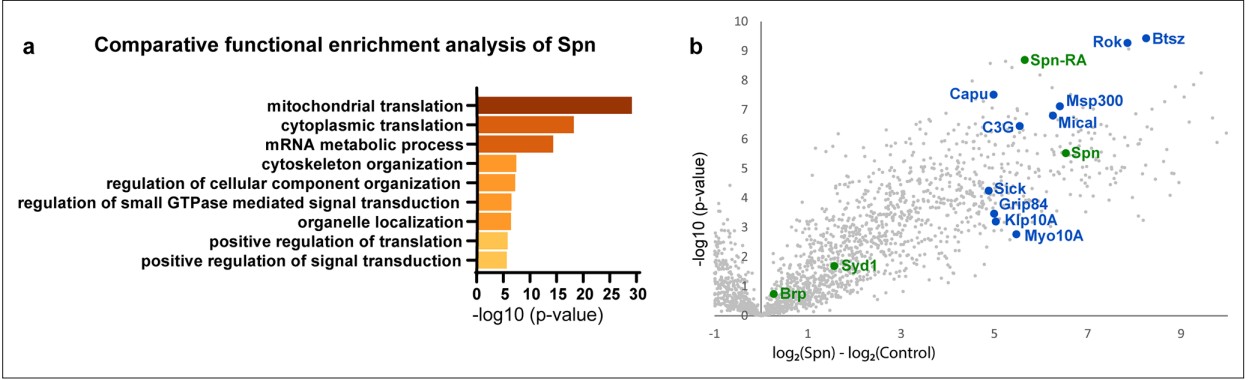

**Figure 3.** Spinophilin co-immunoprecipitates cytoskeleton remodelers. (**a**) Comparative functional enrichment analysis of Spn interactomes suggested that Spn was specifically associated with proteins involved in translation control and cytoskeleton organization. (**b**) Spn coimmunoprecipitated various regulators of F-actin dynamics. Also see *Figure 3—figure supplement 1* for influence of translation on PhTx-induced PHP.

The online version of this article includes the following source data and figure supplement(s) for figure 3:

**Figure supplement 1.** Cycloheximide, a translation elongation blocker, does not affect homeostatic plasticity.

**Figure supplement 1—source data 1.** Table containing exact values for the data depicted in *Figure 3—figure supplement 1*, along with details of statistical analyses.

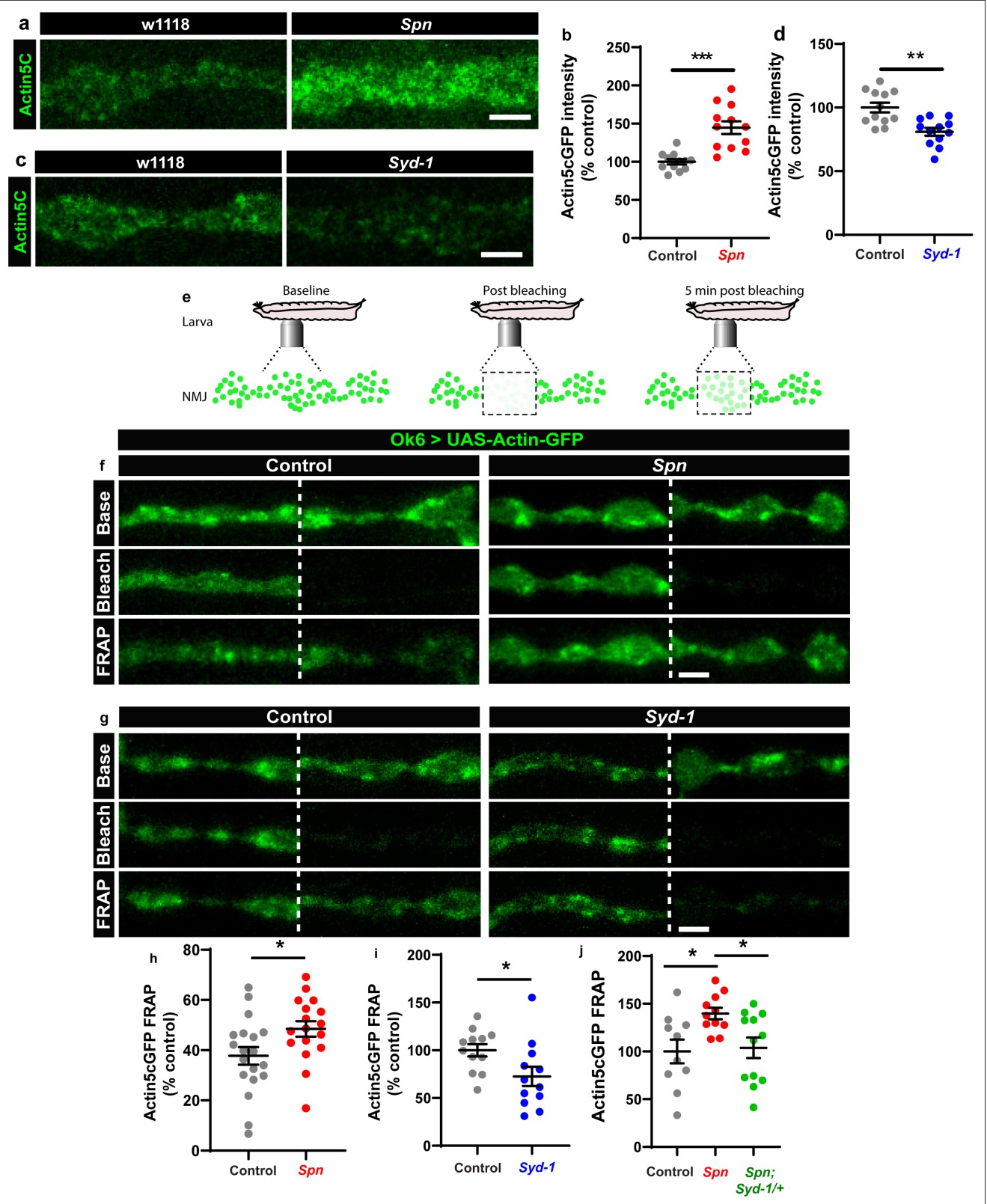

**Figure 4.** The Spinophilin/Syd-1 antagonism converges on presynaptic F-actin dynamics. (**a,c**) Representative images of third-instar larval muscle 4 NMJs showing intrinsic Actin-GFP. Scale bars: 2 μm. (**a–d**) *Spn* mutants show an increase in Actin-GFP intensity while *Syd-1* mutants show a decrease compared to controls. (**b,d**) mean actin-GFP intensities measured through confocal imaging. (**e**) Scheme depicting in vivo fluorescence recovery after photobleaching (FRAP) imaging procedure at developing *Drosophila* larval NMJs at baseline (before photobleaching), immediately post bleaching and

*Figure 4 continued on next page*

*Figure 4 continued*

then again after 5min to track recovery of fluorescently labelled protein. (**f–g**) Representative images of muscle 26/27 NMJs labelled with Actin5C$^{GFP}$. Quantification of FRAP shows Actin5C$^{GFP}$ at the NMJ (**h**) increase in *Spn* mutants, (**i**) decrease in *Syd-1* mutants and (**j**) a rescue of *Spn* mutant phenotype on removal of a single gene copy of Syd-1 (*Spn,Syd-1/+*). Also see *Figure 4—figure supplement 1*. Source data as exact normalized and raw values, detailed statistics including sample sizes and p values are provided in *Figure 4—source data 1*. *p ≤ 0.05; **p ≤ 0.01; ***p ≤ 0.001; n.s., not significant, p > 0.05. All panels show mean ± s.e.m.

The online version of this article includes the following source data and figure supplement(s) for figure 4:

**Source data 1.** Table containing exact values for the data depicted in *Figure 4*, along with details of statistical analyses.

**Figure supplement 1.** Spinophilin and actin colocalize in salivary gland cell cortices.

controls (*Figure 4f and h*). *Syd-1* mutants showed an inverse phenotype with a lower rate of exchange (*Figure 4g and i*). Notably, removing a single gene copy of *syd-1* from *Spn* mutants resulted in a rescue of their actin status (*Figure 4j*).

The *Ok6-Gal4* line also drives expression in salivary gland cells. We observed that Spn and actin were extensively colocalized at the cell cortices of individual salivary gland cells (*Figure 4—figure supplement 1a*). Furthermore, *Spn* mutants specifically showed excessive actin accumulation/ stabilization within cell bodies, which were absent in control animals (*Figure 4—figure supplement 1b*). Thus, our proteomics and FRAP experiments identified local AZ actin dynamics as a possible downstream candidate process for how Spn and Syd-1 antagonistically mediate AZ remodeling and homeostatic plasticity.

## Rescue of *Spinophilin* plasticity deficits after genetic and pharmacological disruption of F-actin

Filamentous actin has been suggested to act as a barrier to vesicle exocytosis (*Aunis and Bader, 1988*). In particular, at the lamprey giant synapse, selective stabilization of the cortical actin pool blocks synaptic transmission by blocking exocytosis (*Bleckert et al., 2012*). Sema2b-PlexB signaling mediates PHP via the evolutionary conserved Mical and disassembles actin filaments and inhibits actin polymerization by specifically oxidizing actin (*Wu et al., 2018*; *Orr et al., 2017*). Mical might function to destabilize plasma membrane-close actin and facilitate vesicle release (*Orr et al., 2022*). Actin depolymerization was also able to independently elevate BRP levels at the *Drosophila* NMJ (*Böhme et al., 2019*).

Based on our results described above, we asked whether *Spn* mutants might suffer defective AZ plasticity maintenance due to having excessive AZ-close F-actin. In Spn IPs, Mical was specifically and highly enriched (*Figure 3b*). *Spn* mutants displayed lower levels of Mical in NMJ immunostainings (*Figure 5—figure supplement 1a–b*), while *Syd-1* mutants showed normal levels of Mical (*Figure 5—figure supplement 1a–b*). Interestingly, NMJ Mical levels are also subject to Spn/Syd-1 antagonism. *Syd-1* heterozygosity in *Spn* mutants resulted in a reestablishment of Mical levels comparable to wild-type NMJs (*Figure 5—figure supplement 1c–d*). We thus asked whether Mical might indeed function downstream of Spn to mediate PHP by depolymerizing actin. Importantly, *Mical* mutants have also previously been found to show defective homeostatic plasticity (*Orr et al., 2017*). We therefore speculated that overexpressing Mical in *Spn* mutants could resolve the excessive actin stabilization seen in *Spn* mutants. Indeed, overexpression of Mical in the motor neurons of *Spn* mutants allowed for BRP increase upon PhTx treatment, comparable to controls (*Figure 5a–b*), and also reestablished 30 min PHP in these animals in terms of QC increase (*Figure 5c and f*) calculated from mini (*Figure 5c and d*) and evoked amplitudes (*Figure 5c and e*). On the contrary, overexpression of a redox-dead version of Mical (*Hung et al., 2011*) did not allow for BRP increase in *Spn* mutants (*Figure 5a–b*) and did not reestablish PHP maintenance here (*Figure 5c–f*). This suggests that Mical has a function downstream of Spn in depolymerizing actin, consequently allowing access to the AZ for longer-lasting structural and functional plasticity.

To independently test the role of actin (de-)polymerization in a more acute fashion, we asked whether blocking actin polymerization with Latrunculin-B (LatB) might be able to reestablish PHP in *Spn* mutants. We found this to indeed be the case (*Figure 5g–l*). Pretreating *Spn* mutants with LatB allowed for PhTx-triggered increase in BRP levels in *Spn* mutants, similar to controls (*Figure 5g–h*). LatB pretreatment also allowed for QC increase (*Figure 5l and l*), calculated from reduced mini (*Figure 5l*

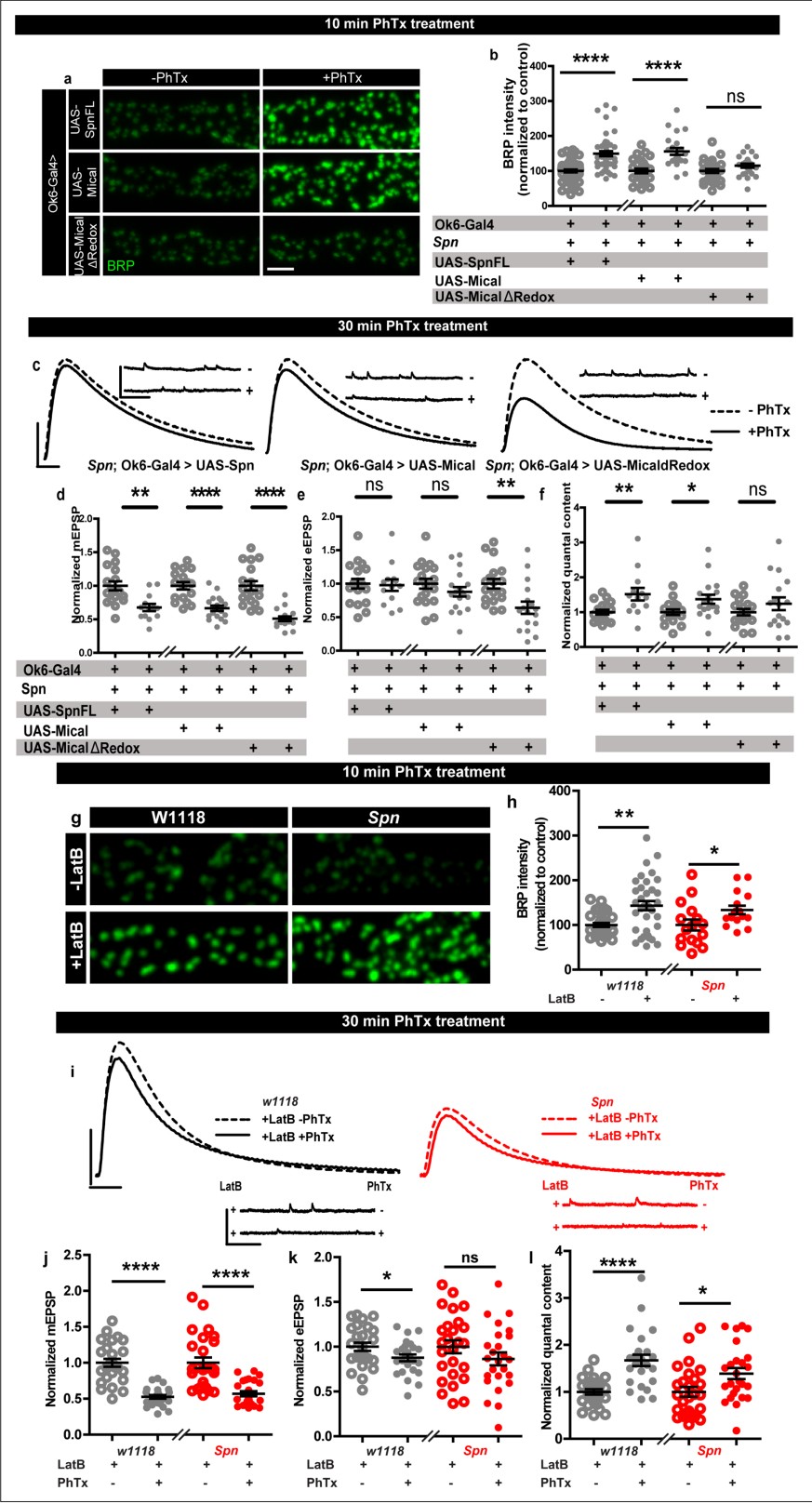

**Figure 5.** Rescue of Spinophilin plasticity deficits after genetic and pharmacological disruption of F-actin.
(**a,g**) Representative images of third-instar larval muscle 4 NMJs immunostained with an antibody against BRP.
Scale bars: 2 μm. (**a–b**) motoneuronal expression of full-length Spn and Mical in *Spn* mutant background reinstates
BRP increase upon PhTx treatment, while MicalΔRedox expression does not. (**g–h**) Latrunculin-B (LatB) treatment

*Figure 5 continued*

results in an increase in BRP in both control and *Spn* mutants. (**B,H**) mean BRP intensities measured through confocal imaging. (**c,i**) Representative traces of mEJP and eEJP measurements at third-instar larval muscle 6/7 NMJs. Scale bars: eEJP, 10ms, 10 mV; mEJP traces 500ms, 5 mV. (**d,j**) mEJP amplitudes are reduced upon PhTx treatment. eEJP amplitudes are compensated (**e**) and QC is increased (**f**) in motoneuronal expression of full-length Spn and Mical in *Spn* mutant background upon 30 min PhTx treatment but not in MicalΔRedox. eEJP amplitudes are compensated (**k**) and QC is increased (**l**) in control animals and *Spn* mutants treated with LatB. Also see *Figure 5—figure supplement 1*. Source data as exact normalized and raw values, detailed statistics including sample sizes and p values are provided in *Figure 5—source data 1*. *p≤0.05; **p≤0.01; ***p≤0.001; n.s., not significant, p>0.05. All panels show mean ±s.e.m.

The online version of this article includes the following source data and figure supplement(s) for figure 5:

**Source data 1.** Table containing exact values for the data depicted in *Figure 5*, along with details of statistical analyses.

**Figure supplement 1.** Actin depolymerization via Mical could underlie Spinophilin-mediated homeostatic plasticity.

**Figure supplement 1—source data 1.** Table containing exact values for the data depicted in *Figure 5—figure supplement 1*, along with details of statistical analyses.

*and j*) and evoked amplitudes (*Figure 5l and k*) during the maintenance phase of PHP. Concisely, *Spn* mutants underwent PHP even at 30 min of PhTx treatment in the presence of LatB. Thus, local, AZ-close pools of cortical F-actin might normally be negatively controlled by Spn, and disrupting the F-actin downstream of the Spn control point is seemingly able to overwrite Spn's absence.

## Plastic increases in release-ready vesicle pool size are facilitated by Spinophilin

We went on to explore the possible consequences of Spn-mediated plasticity associated with F-actin status. Notably, RRP increases contribute to PHP response (*Weyhersmüller et al., 2011*; *Müller et al., 2012*), and F-actin in turn controls RRP size and replenishment (*Wu and Chan, 2022*). Spn might therefore mediate PHP through RRP replenishment and/or maintenance. To address whether *Spn* mutants had altered RRP size compared to controls during PHP, we performed two electrode voltage clamp (TEVC) recordings during high frequency stimulation (*Schneggenburger et al., 1999*; *Hallermann et al., 2010*). *Spn* mutants indeed showed atypically low RRP sizes and SV refilling rates (*Figure 6—figure supplement 1a*). Importantly, while RRP sizes clearly increased in wildtype animals under 30 min PhTx, in *Spn* mutants the RRP size did not increase (*Figure 6a–b*). At the same time, however, RRP refilling rates changed similarly in *Spn* mutants and wildtype under PhTx treatment (*Figure 6a and c*). Thus, Spn seemingly is needed to allow for plastic RRP size increases, which coincides with *Spn* mutant's inability to upregulate BRP and Unc13A levels upon PhTx treatment (*Figure 1a–d*).

Given these alterations in RRP, we went on asking whether the SV distribution was altered in *Spn* mutants. Synaptotagmin-1 (Syt1) levels measured over the entire NMJ area showed no difference when comparing wildtype and *Spn* mutants (*Figure 6—figure supplement 1d–e*) suggesting that the total number of SVs is unchanged in *Spn* mutants. Indeed, transmission electron microscopic analysis of NMJ boutons showed that the overall number of SVs and the overall SV density (number of SVs/unit area) were essentially unaltered in *Spn* mutants (*Figure 6d–f*). At the same time, however, in the vicinity of the AZs (recognized by their electron dense T-bars), SV numbers appeared clearly reduced in *Spn* mutants (*Figure 6d and g*). Indeed, the SV micro-arrangement was apparently altered at AZs, as the SVs no longer clustered at T-bars (*Figure 6d*). Counting SVs in increments of 20 nm shells around the T-bar showed that the number of SVs very close (<110 nm from T-bar center) to the AZs was lower in *Spn* mutants, while further away (>110 nm from T-bar center), SV densities were normal (*Figure 6d and h*). Thus, SVs in *Spn* mutants seem to be largely excluded from the area very close to the AZ membrane, arguably explained by excessive cortical actin filaments blocking AZ access in *Spn* mutants (see discussion).

As Spn seems to be involved in RRP size maintenance, we looked for possible additional mechanisms that could be involved in this context. Our proteomics analysis identified Rho-associated protein kinase (ROK) as strongly enriched within the Spn-IP (*Figure 3b*). Rok has been shown to be involved in actomyosin contraction, RRP maintenance and facilitating SV docking (*González-Forero et al., 2012*).

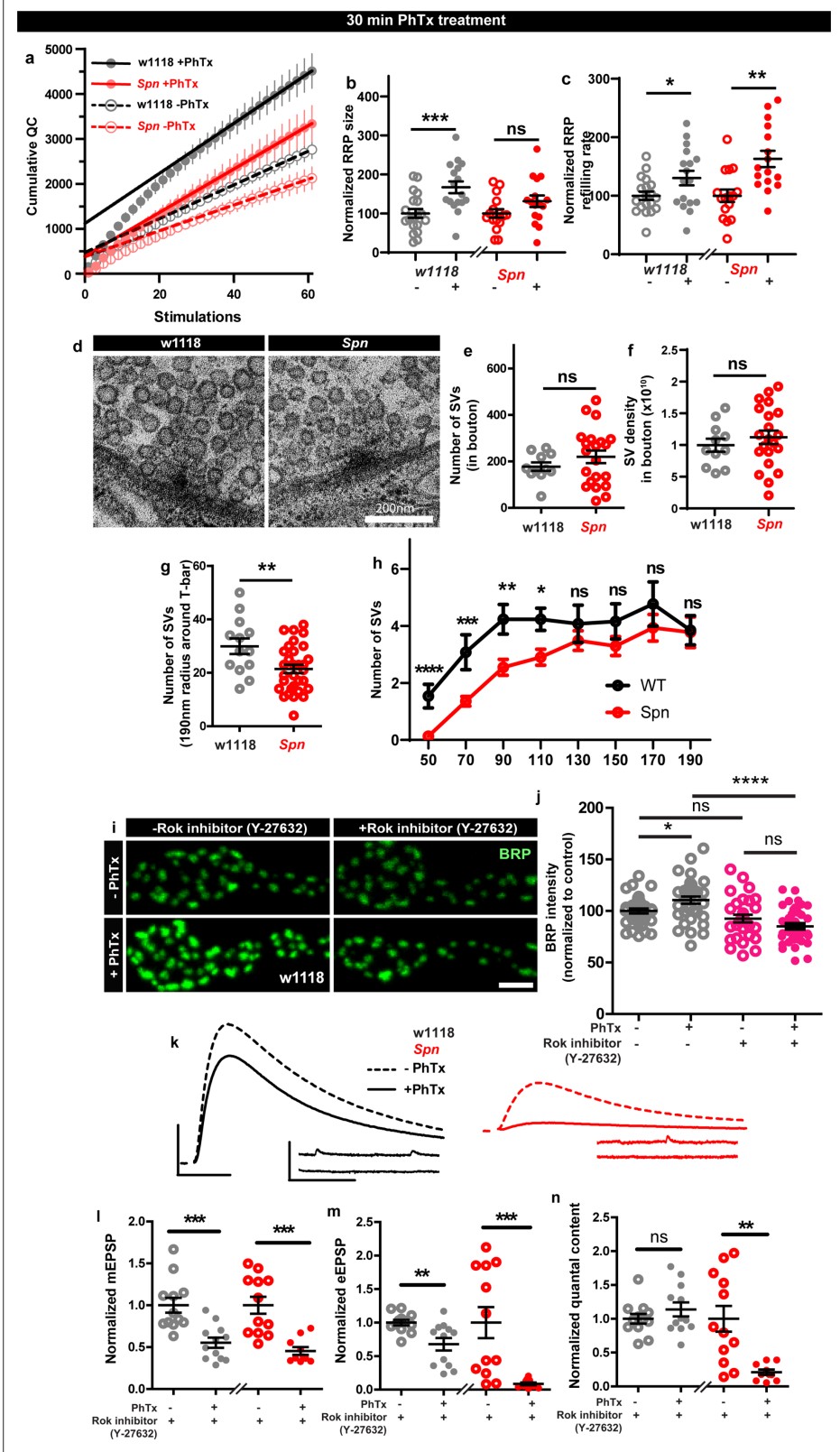

**Figure 6.** Spinophilin facilitates plastic increases in release-ready vesicle pool size. Quantification of the RRP size (**b**) and RRP refilling rate (**c**) from (**a**) Average cumulative quantal content plotted against the number of stimulations with linear regression of the steady state amplitudes. (**d**) Representative electron microscopy images of control showing normal SV distribution and *Spn* mutants showing sparse SV distribution around the AZ T-bar.

*Figure 6 continued on next page*

*Figure 6 continued*

*Spn* mutants have a normal number of SVs in the bouton quantified in (**e**) absolute number of SVs and (**f**) SV density in bouton, but have (**g**) fewer SVs in the vicinity of the AZs. (**h**) SVs distribution at increasing distance from the center of the AZ shows that *Spn* mutants have fewer SVs near the AZ center, but have normal numbers of SVs further away from the AZ center. (**i**) Representative images of third-instar larval muscle 4 NMJs immunostained with an antibody against BRP. Scale bars: 2 μm. (**j**) Preincubating larvae with Rok inhibitor results in their inability to upregulate BRP upon PhTx treatment. (**k**) Representative traces of mEJP and eEJP measurements at third-instar larval muscle 6/7 NMJs. Scale bars: eEJP, 10ms, 10 mV; mEJP traces 500ms, 5 mV. (**l**) mEJP amplitudes are reduced upommican PhTx treatment. Upon PhTx treatment, eEJP amplitudes are not compensated (**m**) and QC does not increase (**n**) in Rok-inhibitor treated controls and *Spn* mutants. Also see *Figure 6—figure supplement 1*. Source data as exact normalized and raw values, detailed statistics including sample sizes and p values are provided in *Figure 6—source data 1*. *p≤0.05; **p≤0.01; ***p≤0.001; n.s., not significant, p>0.05. All panels show mean ±s.e.m.

The online version of this article includes the following source data and figure supplement(s) for figure 6:

**Source data 1.** Table containing exact values for the data depicted in *Figure 6*, along with details of statistical analyses.

**Figure supplement 1.** Spinophilin facilitates RRP increase during sustained homeostatic plasticity.

**Figure supplement 1—source data 1.** Table containing exact values for the data depicted in *Figure 6—figure supplement 1*, along with details of statistical analyses.

---

We therefore tested whether Rok inhibition using a specific inhibitor could preclude PHP maintenance. Pretreating larvae with Rok inhibitor did indeed preclude BRP increases upon PhTx treatment (*Figure 6i–j*). Cownsistently, the larvae pretreated with the Rok inhibitor also could not compensate for PhTx challenge as they did not increase their QC (*Figure 6n*). Control and *Spn* mutants, pretreated with Rok inhibitor, showed a reduction in mini amplitudes with PhTx treatment (*Figure 6kl*), but the Rok inhibitor pretreated wildtype controls did not return their evoked release to baseline (*Figure 6k and m*). Notably, at the same time, inhibitor treatment did not affect baseline release activity (*Figure 5—figure supplement 1i–l*), suggesting that actomyosin function is indeed needed for fast, plastic RRP expansion in our system, and might be downstream of Spn.

## Spinophilin is crucial for aversive olfactory mid-term memories but not for learning

The *Drosophila* mushroom body (MB) is a leading system for the analysis of learning and subsequent memory formation and consolidation. Within the MB principal neurons (Kenyon cells, KCs), presynaptic plasticity in response to conditioning an odor with contextual information (electric shock in the case of aversive conditioning) is meant to be the basis for learning but also subsequent memory formation and consolidation (*Krashes et al., 2007*; *de Belle and Heisenberg, 1994*). Notably, our recent work showed that BRP and Unc13A levels in the MB increased over about 3 hr after conditioning to later return to baseline levels (*Turrel et al., 2022*), consistent with previous work (*Zhang et al., 2018*) reporting a transient, conditioning induced BRP increase. As BRP increases were also transient under PhTx at the NMJ as well (*Figure 2a–b and m–n*), we wondered whether Spn within KCs would be of importance for conditioning-triggered increases of BRP and Unc13A. Thus, we stained MBs of wild type animals subjected to olfactory conditioning for BRP and Unc13A. As expected (*Turrel et al., 2022*), we observed an upregulation of BRP and Unc13A in response to paired conditioning after both 1 and 3 hr post conditioning in comparison to naive untrained animals (*Figure 7a and c*). We analogously knocked down Spn specifically in the MB KCs post-developmentally, starting from 3 days after hatching, by combining the temperature-sensitive Gal80 inhibitor (Gal80ts) with RNA interference (RNAi) via the TARGET system using Gal80ts to block Gal4 transcriptional activity at low temperature (18 °C). At high temperature (29 °C), the Gal80ts protein is denatured and consequently the inhibition of Gal4 activity is lifted. In order to restrict RNAi expression to adult MB lobes, we combined tub- Gal80ts with the KC generic OK107-Gal4 line (Gal80ts;OK107). Notably, the post-conditioning increases of BRP and Unc13A did not occur in the Spn-directed post-developmental KC-specific knockdown (KD) situation (*Figure 7b and d*), while per se the BRP and neuropil organization appeared unchanged in these animals as compared to isogenic controls on the confocal analysis level (data now shown).

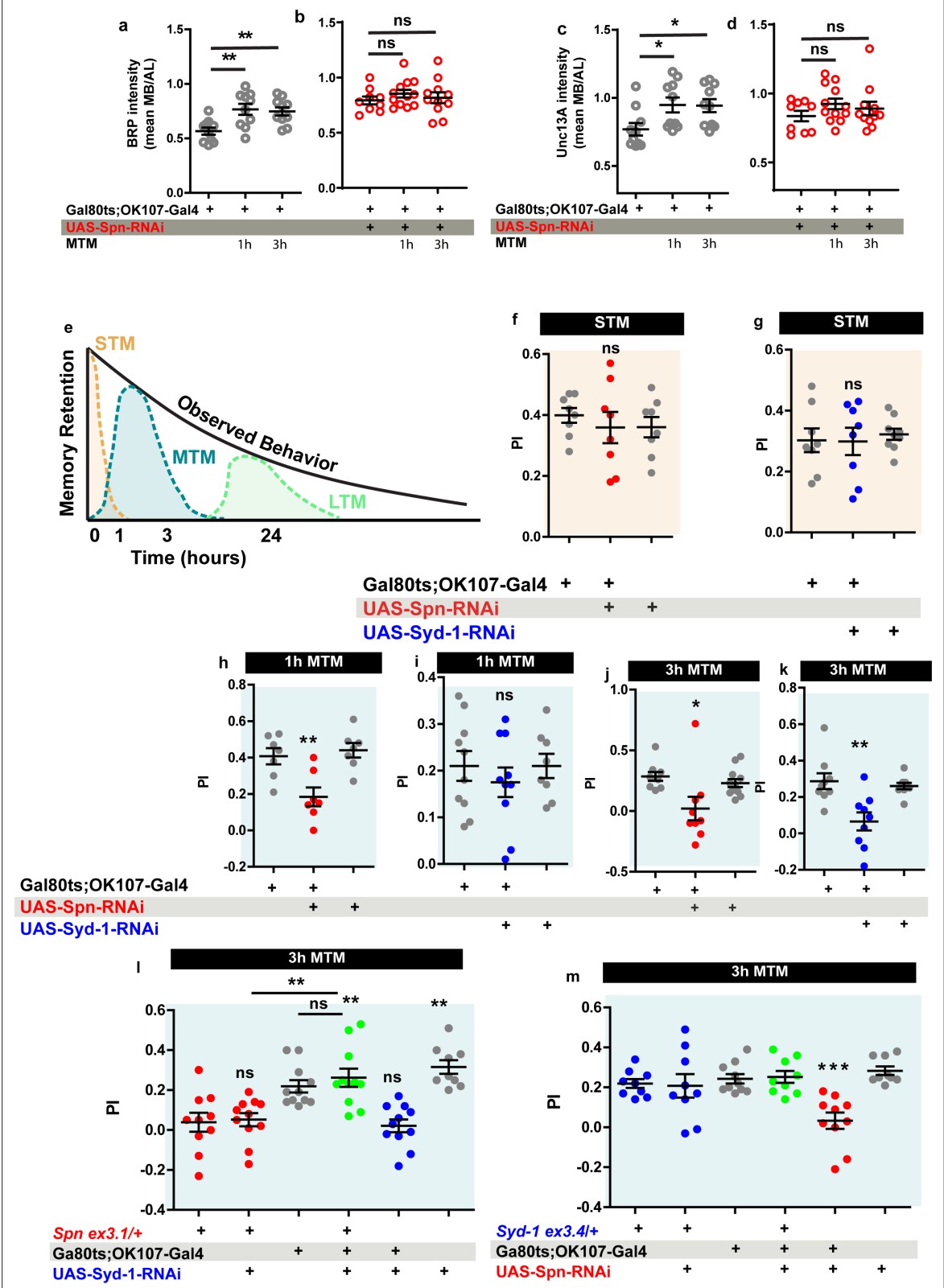

**Figure 7.** Spinophilin is crucial for aversive olfactory mid-term memories but not for learning. The Spinophilin/Syd-1 antagonism controls aversive mid-term memories. (**a–d**) Quantification of BRP and Unc13A intensity in control and *Spn* knockdown (KD) flies, 1 and 3 h after conditioning. (**a,c**) BRP and Unc13A levels increase in controls upon paired conditioning while (**b,d**) BRP and Unc13A do not increase in Spn KD upon paired conditioning. (**e**) Scheme showing different phases of memory in *Drosophila*. (**f–g**) STM is normal in both Spn and Syd-1 KDs. (**h**) 1 hour and (**j**) 3 hour MTM are both

*Figure 7 continued*

impaired in Spn KD. (**i**) 1 hour MTM is normal while (**k**) 3 hour MTM is impaired in *Syd-1* KD. (**l**) *Spn* heterozygosity (*Spn*^ex3.1^/+) shows impaired 3 hr MTM, which is rescued upon concomitant KD of Syd-1. (**m**) 3 hour MTM impairment in Spn KD is rescued by concomitant reduction in Syd-1 through *Syd-1* heterozygosity (*Syd-1*^ex3.4^/+). Also see ***Figure 7—figure supplements 1–3***. Source data as exact normalized and raw values, detailed statistics including sample sizes and p values are provided in ***Figure 7—source data 1***. *p≤0.05; **p≤0.01; ***p≤0.001; n.s., not significant, p>0.05. All panels show mean ±s.e.m.

The online version of this article includes the following source data and figure supplement(s) for figure 7:

**Source data 1.** Table containing exact values for the data depicted in ***Figure 7***, along with details of statistical analyses.

**Figure supplement 1.** Spinophilin and Syd-1 knockdown results in impaired mid-term memory (MTM), due to impaired anesthesia-sensitive memory (ASM).

**Figure supplement 1—source data 1.** Table containining exact values for the data depicted in ***Figure 7—figure supplement 1***, along with details of statistical analyses.

**Figure supplement 2.** The Spinophilin/Syd-1 antagonism is not relevant to 1h mid-term memory (MTM) formation.

**Figure supplement 2—source data 1.** Table containining exact values for the data depicted in ***Figure 7—figure supplement 2***, along with details of statistical analyses.

**Figure supplement 3.** Model depicting plasticity processes underlying presynaptic homeostatic plasticity in peripheral neuromuscular synapses and memory stabilization in mushroom body KCs.

Specifically triggering post-developmental KD of BRP in KCs drastically reduces aversive mid-term memory (MTM) measured at 1 or 3 hr after conditioning (***Turrel et al., 2022***). We thus tested the Spn-KD animals (***Figure 7e***) for aversive short-term memory (STM, measured 5 minutes after conditioning) and MTM (1 or 3 hr after conditioning). Importantly, KC-specific Spn KD in adults did not result in any alteration of STM scores (***Figure 7f***). In other words, initial learning was intact here, consistent with the principal synaptic architecture and circuitry not being majorly affected by our post-developmental manipulation. At the same time, however, both the 1 hr (***Figure 7h***) and 3 hr (***Figure 7j***) MTM scores were significantly and strongly reduced.

Olfactory aversive mid-term memory can be further dissected into anesthesia-sensitive memory (ASM) and anesthesia-resistant memory (ARM) components, which also differ concerning their molecular underpinnings (***Quinn and Dudai, 1976***; ***Scheunemann et al., 2012***). We recently showed that presynaptic AZ plasticity is specifically affecting ASM but not ARM (and neither STM/learning) (***Turrel et al., 2022***). Consistent with Spn being upstream of BRP-mediated plasticity, Spn-KD also specifically reduced the ASM component but left ARM intact (***Figure 7—figure supplement 1c–d,f-g***). Control experiments conducted in animals maintained at 18 °C showed no memory defects (***Figure 7—figure supplement 1e and h***). In short, Spn-mediated plasticity likely plays a crucial role for stabilizing memories once initial learning has been accomplished. We next asked whether sustained AZ plasticity operated according to a mechanistically conserved set of rules across different *Drosophila* synapses. To this end, we went on to ask whether the Spn/Syd-1 antagonism, functioning at NMJ synapse, could be involved in MTM formation as well.

## The Spinophilin/Syd-1 antagonism controls aversive mid-term memories

We first tested for the consequences of post-developmental KC-specific Syd-1 KD and found that STM (***Figure 7g***) remained intact. Concerning 1 hr MTM, Syd-1 KD did not trigger significant reduction of memory scores (***Figure 7i***). For 3 hr MTM, however, scores were significantly reduced (***Figure 7k***). Once again, this deficit was due to an impairment in the ASM component (***Figure 7—figure supplement 1i–j***). Respective 18 °C control experiments did not return any memory deficits (***Figure 7—figure supplement 1k***).

We then tested for a genetic interaction between Spn and Syd-1 in MTM formation. We found that reducing *spn* to a single copy (using an excision null allele in heterozygosity, *Spn*^Ex3.1^/+) similarly reduced MTM scores, as did Spn-RNAi KD (***Figure 7h, j and l*** red). Thus, aversive ASM formation appears highly sensitive to a moderate reduction of Spn protein levels (please note that *Spn* null mutants are not adult viable and could not be used for memory experimentation).

While at the NMJ, *Spn* and *Syd-1* show opposite phenotypes during PhTx-induced plasticity, the reduction of both Spn and Syd-1, individually reduced MTM scores. We suspected that plasticity had to operate in a defined dynamic range in order to allow for proper encoding of information, with

both synaptic hypo- and hyper-plasticity possibly rendering synaptic information storage unproductive. Indeed, concomitant RNAi mediated KD of Syd-1 in MB neurons could specifically reestablish 3 hr MTM in *Spn* heterozygosity (*Figure 7l*). Importantly, control experiments at 18 °C did not show this suppression (*Figure 7—figure supplement 2c*). In contrast, at 1 hr MTM, there was no rescue of *Spn* mutants by concomitant Syd-1 KD (*Figure 7—figure supplement 2a*), consistent with Syd-1 not affecting 1 hr MTM (*Figure 7l*).

To independently validate that Spn/Syd-1 antagonism steers MTM formation, we reversed the genetic strategy by combining a Spn KD with *Syd-1* heterozygosity using an excision null allele (Syd-1$^{Ex3.4}$/+). Importantly, this resulted in comparable results. *Syd-1* heterozygosity as such did not render a memory phenotype (*Figure 7—figure supplement 2b*, *Figure 7m*). As expected, MB-specific Spn KD showed impaired 1 hr and 3 hr MTM (*Figure 7—figure supplement 2b*, *Figure 7m*). Again, *syd-1* heterozygosity could specifically rescue 3 hr MTM (*Figure 7m*) in *Spn* KD but not 1 hr MTM (*Figure 7—figure supplement 2a*). Control experiments conducted in animals maintained at 18 °C again showed no memory defects (*Figure 7—figure supplement 2d*). Thus, two independent experiments suggest that indeed a Spn/Syd-1 antagonism is also involved in the longer component of MTM in the MBs.

## Discussion

Acute formation of memory engrams and subsequent refinement processes likely entail changes of both the postsynaptic as well as the presynaptic compartments, and involves both Hebbian as well as homeostatic forms of synaptic plasticity. In this regard, we recently showed that in the *Drosophila* brain, AZ remodeling (detected by level changes of ELKS-family AZ scaffold protein BRP and munc13-1 family member Unc13A) takes place within MB intrinsic neurons (Kenyon cells, KCs) after paired conditioning over a time window of few hours. Notably, we provided genetic evidence that this AZ remodeling within the KCs is specifically needed for mid-term aversive olfactory memories (*Turrel et al., 2022*). Here, motivated by the generically high level of evolutionary conservation of presynaptic release machinery, we further analyzed regulatory processes controlling presynaptic homeostatic potentiation (PHP) to then address their role in MB-mediated memory formation.

### An equilibrium between Spn and Syd-1 regulation controls the maintenance phase of NMJ homeostatic active zone plasticity

Homeostatic plasticity is a collective term for compensatory physiological processes that counterbalance neuronal perturbations, ensuring brain stability. Recently, it was shown that also in the adult rodent hippocampus, homeostatic potentiation of presynaptic neurotransmitter release compensates acute, partial blockade of postsynaptic GluRs (*Chipman et al., 2022*). This finding is consistent with the expression of PHP documented at peripheral synapses, prominently at the *Drosophila* larval NMJ synapse. The proteins and mechanisms involved in PHP have been shown to extend their functionality to mid-term memory formation in the *Drosophila* central synapses, as shown in this study involving Spn/Syd-1, and in previous studies involving BRP, Arl8 and IMAC (*Turrel et al., 2022*).

At NMJ synapses, a spectrum of signaling molecules and pathways promote homeostatic plasticity, including BMP signaling, CaMKII signaling, TOR signaling, proteasomal degradation, and transsynaptic signaling (*Davis and Müller, 2015*; *Frank, 2014*; *Orr et al., 2017*; *Frank et al., 2006*). These factors enhance presynaptic transmitter release via increasing the number of voltage-gated Ca$^{2+}$ channels and consequently enhancing Ca$^{2+}$ influx, as well as by increasing the number of release sites or release-ready SVs. Structural AZ-remodeling occurs within minutes, but upstream regulatory mechanisms triggering and coordinating AZ remodeling have scarcely been described (*Frank et al., 2020*). Our data now suggest that presynaptic Spn operates high in a regulatory plasticity hierarchy controlling AZ scaffold remodeling to provide additional release sites and increase the number of release-ready SVs. Although AZ-remodeling (*Böhme et al., 2019*; *Goel et al., 2017*; *Weyhersmüller et al., 2011*; *Gratz et al., 2019*) begins within a few minutes after PHP induction (via PhTx), it is not a prerequisite for the rapid increase of QC observed. Concretely, *BRP*, *Aplip-1*, *Srpk* (*Böhme et al., 2019*), *Arl8* and *IMAC* mutants (*Turrel et al., 2022*), which are all deficient in AZ scaffold remodeling, still increase QC in the first few minutes after the PhTx challenge (PHP induction). Later, however, they are unable to sustain a QC increase as early as 30 min after PhTx treatment.

We suggest that the transient increase of BRP, also previously described specifically in the MB γ-neurons (*Zhang et al., 2018*), triggers other, longer lasting AZ changes. Indeed, we found that the increase of the critical release factor Unc13A is still present at 30 min PhTx treatment and is dependent on the "transient" BRP increase (Fig. S3B) (*Turrel et al., 2022*). *Turrel et al., 2022* also uncovered a more transient upregulation of BRP when compared to Unc13A in the MB. Here, specifically upon paired olfactory conditioning, 1 hr after training, animals displayed BRP and Unc13A level increases. At 3 hr post training, however, BRP levels had already plateaued, whereas Unc13A levels had increased further (*Turrel et al., 2022*). During the induction phase, QC increase seemingly is the result of maximizing the output of already existing release machinery and release sites via increased $Ca^{2+}$-influx or activation of preexisting but dormant release sites in a RIM, RIM-BP and Unc13A dependent fashion. Sustaining QC enhancement over a longer period, however, seemingly requires structural AZ remodeling. Arguably, the induction phase might trigger the maintenance phase, given that several mutants were found to induce but not sustain PHP (*Böhme et al., 2019*; *Turrel et al., 2022*; *Frank et al., 2009*; *Marie et al., 2010*), while the reverse case has not been described so far. We show that Spn-mediated BRP-incorporation, triggered during the induction phase, is essential for PHP maintenance and likely functions to increase the number of SV release sites at the AZ by installing more Unc13A. Like BRP itself, Spn is per se dispensable for PHP induction in the tested paradigms. This said, however, a relatively bigger variance in the evoked release responses of *Spn* mutants was observed (*Figure 1h*), indicating that *Spn* mutants are not completely unaffected also concerning the induction phase of homeostatic plasticity. Induction and maintenance mechanisms might indeed be linked in more complex ways, which remain to be worked out in the future. PHP in *Spn* mutants could be reinstated by attenuating Syd-1 levels by removing a single *syd-1* gene copy, extending the developmental role of Spn/Syd-1 antagonism into PHP. Interestingly, Spn and Syd-1 also bind each other as shown through yeast-2-hybrid assay (*Muhammad et al., 2015*) and co-immunoprecipitation (this study). Future work could lead to an understanding on how their physical association steers their complementary functions.

## Local, active zone close control of F-actin pools downstream of Spn/Syd-1 controlled remodeling

While AZ remodeling has been described in terms of changes in core AZ proteins during PHP, there is much to be learned concerning the regulatory proteins which mediate sustained AZ remodeling during PHP (*Frank et al., 2020*). The actin-based cytoskeleton has been shown to support RRP recruitment, docking and recycling of SVs after neurotransmitter release (*Cingolani and Goda, 2008*; *Hallermann and Silver, 2013*; *Rust and Maritzen, 2015*; *Miki et al., 2016*). Actomyosin contraction has also been implicated in the translocation of SVs from the reserve pool to the presynaptic plasma membrane. In both Xenopus nerve-muscle cocultures and the *Drosophila* NMJ, disruption of actin filaments decreases the depression of neurotransmitter release that ordinarily occurs during high-frequency stimulation (*Kuromi and Kidokoro, 1998*; *Wang et al., 1996*). Actin remodeling has also been implicated in memory formation in different model organisms (*Lamprecht, 2016*; *Kramár et al., 2006*; *Krucker et al., 2000*; *Frambach et al., 2004*; *Ganeshina et al., 2012*).

Our previous work showed that acute actin-depolymerization via LatB prevents the PhTx-induced BRP increases at AZs (*Böhme et al., 2019*). Moreover, *Drosophila* Mical, a highly conserved, multi-domain cytoplasmic protein that mediates actin depolymerization, was previously shown to be necessary for PHP (*Orr et al., 2017*). Actin disassembly has been described before in the release of large vesicles from chromaffin cells (*Rosé et al., 2001*), and also more recently at the AZ via Mical to facilitate vesicle fusion at the AZ membrane (*Orr et al., 2022*). Motivated by Spn-IPs containing Mical and by the established roles of Spn family proteins in regulating actin (*Chia et al., 2012*; *Nakanishi et al., 1997*; *Ryan et al., 2005*), we tested whether indeed actin dynamics were mechanistically downstream of Spn and Syd-1 here. Indeed, our data suggest that the Spn/Syd-1 antagonism converges on controlling local AZ-close F-actin abundance and dynamics. Spn/Syd-1 regulation of F-actin status in turn seems to control the accessibility of SV at their release sites. To what extent the modulation of local F-actin promoting RRP size and SV docking might directly couple to the accumulation of more AZ scaffold material, potentially via biosynthetic precursor vesicles (*Vukoja et al., 2018*), is an interesting question now. Recent work suggests two pools of presynaptic actin to be involved in different functions. Here, one F-actin pool within the SV field facing the AZ scaffold might be involved in vesicle

recruitment while another pool might operate close to the AZ plasma membrane and control RRP size and vesicle fusion (*Orr et al., 2022*). We here show evidence that the AZ remodeling and PHP maintenance process is mediated by cortical actin disassembly at the AZ via Mical, likely positioned near the AZ by Spn (*Figure 7—figure supplement 3*, Spn-dependent cortical actin depolymerization).

We recently found that PHP maintenance is driven via a compaction process involving both voltage-gated $Ca^{2+}$ channel Cacophony and BRP along with increases in the number of $Ca^{2+}$ channel molecules at the AZ center (*Ghelani et al., 2023*). Such a compaction process could be mediated by local depolymerization of F-actin as implicated by our results (*Mrestani et al., 2021*) and also facilitate increases of AZ scaffold protein numbers (depicted in *Figure 7—figure supplement 3*). Notably, during tight junction formation, actin filament depolymerization by latrunculin-A in polarized MDCK cells led to the formation of droplet-like puncta of Zonula Occludens-1 through liquid-liquid phase separation (LLPS) (*Beutel et al., 2019*; *Schwayer et al., 2019*). LLPS has been described in various contexts as forming membraneless protein condensates and by increasing local protein concentrations. Notably, in *C. elegans*, Syd-2 and ELKS-1 (BRP homologue) were shown to form liquid phases at developing AZs, which finally mature into more stable structures. Condensate liquidity is suggested to promote the incorporation and mixing of AZ components (*McDonald et al., 2020*). PHP at AZs could thus involve LLPS-like processes involved in the concentration and accumulation of liquid-phase promoting AZ proteins driven by Spn-mediated F-actin depolymerization (*Figure 7—figure supplement 3*, AZ contraction).

We also found that Rok kinase (found in Spn IPs) precludes PHP maintenance, possibly by restricting RRP replenishment (*Figure 6I–J*, *Figure 7—figure supplement 3*, Rok-dependent RRP replenishment). Basally active ROCK (*Drosophila* Rok) was found to inhibit actomyosin contractility (*González-Forero et al., 2012*) and physiological modulators of ROCK activity have been suggested to trigger short term synaptic plasticity in the nervous system (*González-Forero et al., 2012*). The most studied LLPS process at the AZ is the synapsin condensate (depicted in *Figure 7—figure supplement 3*) which includes SVs and forms the reserve pool (*Milovanovic et al., 2018*), from which Rok-mediated RRP replenishment occurs. Spn likely functions upstream of Rok and Mical in modulating cortical actin to allow access to the release sites, specifically during plasticity processes (*Figure 6a and b*). RRP refilling rates were not affected in *Spn* during plasticity suggesting that recruitment of SVs from the reserve pool is independent of Spn (*Figure 6a and c*). How distinct F-actin remodeling mechanisms in the separate pools of shorter cortical actin filaments and longer filaments further away from the presynaptic membrane (*Orr et al., 2022*) coordinate RRP replenishment and SV docking remain to be investigated. Relevant in this regard, we have shown that the formin DAAM is tightly associated with the synaptic active zone scaffold, and electrophysiological data point to a role in the modulation of SV release (*Migh et al., 2018*).

## A convergence of F-actin dynamics and active zone remodeling in the orchestration of memory relevant presynaptic plasticity

Actin cytoskeleton polymerization during and shortly after learning is needed for long-term memory formation in mammals (*Lamprecht, 2016*; *Kramár et al., 2006*; *Krucker et al., 2000*). Abundant actin filaments have been found in insect MBs and it has been suggested that F-actin remodeling is required for memory formation in insects (*Frambach et al., 2004*). Actin depolymerization enhances associative olfactory memory in the honeybee (*Ganeshina et al., 2012*). Cofilin is an actin severing protein whose constitutive activity is regulated by Rac. In *Drosophila*, inhibition of the Rac signaling pathway stabilizes memory and Rac activation accelerates memory decay (*Shuai et al., 2010*; *Davis, 2010*). In contrast, in mammals, Rac activation promotes memory stabilization (*Rex et al., 2009*). We here show that Spn and Syd-1 are essential in different phases of MTM formation, specifically for the ASM component. Furthermore, we show that Spn function is antagonized by another AZ regulator, Syd-1, and together, they set limits for AZ plasticity both at the NMJ and at central synapses during MTM formation.

# An antagonism between Spinophilin and Syd-1 operates upstream of behaviorally relevant presynaptic long-term plasticity

We here identified an antagonistic mechanism between two evolutionarily conserved regulators, Spn and Syd-1, which controls sustained AZ PHP and at the same time also olfactory memory stabilization in the *Drosophila* learning and memory center.

Although these proteins were described to function in AZ assembly, we are confident that their role in plasticity is independent of their developmental defects for a number of reasons: firstly, the *Spn* PHP phenotype could be rescued by acute application of LatB; secondly, Spn's memory phenotype was tested post-developmentally and could be rescued at the adult stage by lowering Syd-1 levels; thirdly, *Nlg2* mutants which have a very similar developmental phenotype to *Spn* mutants (**Ramesh et al., 2021**) do not show a defect in PhTx-triggered PHP.

The *acute, immediate* formation of aversive short-term memory (STM) was previously shown to trigger synaptic depression at the KC >MBON synapse in the respective MB compartment (**Cohn et al., 2015**; **Hige et al., 2015**; **Owald and Waddell, 2015**). Important for this work, we recently showed that post-developmental BRP KD within the neurons of the adult *Drosophila* MB severely hampered olfactory aversive MTM measured a few hours after conditioning (**Turrel et al., 2022**), while still allowing for proper learning in the minutes range (STM) (**Turrel et al., 2022**). This manuscript shows that Spn KD also does not affect the *acute, immediate* formation of aversive STM, therefore we do not a priori assume that the Spn-mediated AZ remodeling results in synaptic depression. We favor the idea that our plasticity mechanism executes a post-conditioning aftermath critical for consolidating memories over time. This derives from the consideration that synaptic weights, in all likelihood, have to get re-normalized in the MB circuitry after the acute, immediate STM encoding depression. We thus think that one possibility to explain these findings is that the BRP/Unc13A increases that we observe promote potentiation in the MB network to facilitate such a renormalization. Still, the exact relation of the AZ remodeling described in this paper, to STM depression specifically, and KC presynaptic plasticity in general, is currently unknown. What we *can* say, however, is that the molecular mechanisms of sustained presynaptic active zone remodeling, which are accessible via probing homeostatic plasticity at NMJ synapses, are likely of behavioral relevance in *Drosophila*. We here exploited this connection in order to enrich our principle understanding of the synaptic mechanisms driving sustained presynaptic plasticity. This study and **Turrel et al., 2022** provide evidence for an overlap of the executory machinery involved in both NMJ PHP plasticity and MTM formation, as BRP, Spn, Arl8, IMAC and Aplip1 are involved specifically both in mid-term NMJ PHP (at 30 min after PhTx treatment) and in MTM.

Spinophilin and Syd-1 may be pertinent candidates to be tested for long-term plasticity, and their relevant roles in mammalian behavior as well.

## Methods

### Fly strains

*Drosophila melanogaster* strains were maintained as stocks at room temperature. Fly strains for experiments were reared at 25 °C under standard conditions (**Sigrist et al., 2003**) on semi-defined medium (Bloomington recipe). No selection was done based on sex. Genotypes used for experiments were wild type WT: (+/+ (w1118)). *Spn*: (Spn$^{\Delta3.1}$/dfBSc116). *Syd*-1: (Syd-1$^{ex3.4}$/ Syd-1$^{ex1.2}$). *Nrx*-1: (Nrx-1$^{241}$ /Nrx-1df). *Nlg1*: (Nlg1ex$^{2.3}$ /Nlg1$^{ex1.9}$). *Nlg2*: (Nlg2$^{CL5}$/ Nlg2$^{CL5}$). *Spn, Syd*-1+/-: (Spn$^{\Delta3.1}$, Syd-1$^{ex3.4}$/dfBSc116). *Spn, Syd-1+/-*: (Spn$^{\Delta3.1}$, Syd-1$^{ex3.4}$/dfBSc116). Ok6-Gal4>UAS-Actin-GFP: (Ok6-Gal4/UAS-Actin5C-GFP). Ok6-Gal4>UAS-Actin-GFP, *Spn*: (Ok6-Gal4/UAS-Actin5C-GFP; Spn$^{\Delta3.1}$/dfBSc116). Ok6-Gal4>UAS-Actin-GFP, *Syd-1*: (Ok6-Gal4/UAS-Actin5C-GFP; Syd-1$^{ex3.4}$/ Syd-1$^{ex1.2}$). Ok6-Gal4>UAS-Actin-GFP, *Spn;Syd-1/+*: (Ok6-Gal4/UAS-Actin5C-GFP; Spn$^{\Delta3}$.1,Syd-1$^{ex3.4}$/dfBSc116). Ok6-Gal4>UAS Spn, *Spn*: (Ok6-Gal4/UAS-Spn; Spn$^{\Delta3.1}$/dfBSc116). Ok6-Gal4>UAS-Mical, *Spn*: (Ok6-Gal4/UAS-Mical; Spn$^{\Delta3.1}$/dfBSc116). Ok6-Gal4>UAS-MicaldRedox, *Spn*: (Ok6-Gal4/UAS-MicaldRedox; Spn$^{\Delta3.1}$/dfBSc116).

For behavioural experiments, *Drosophila* wild-type strain *w1118* and mutant flies were raised on conventional cornmeal-agar medium in 60% humidity in a 12 hr light/dark cycle at 18 °C. All lines used for memory experiments were outcrossed to the *w1118* background. RNAi stocks were obtained from the Vienna *Drosophila* Resource Center (Austria) for RNAi-Syd1 (VDRC 106241) and Bloomington *Drosophila* Stock Center for RNAi-Spn (Bloomington KK 109888). Syd-1$^{ex3.4}$ and spn$^{\Delta3.1}$ [25]

were combined with RNAi-Spn and RNAi-Syd-1 respectively for the rescue experiments. The *tubulin-Gal80ts;OK107* driver (*Gal80ts;OK107*) was used for conditional expression in the MB. To induce RNAi expression specifically in adults, the TARGET system was used as described by *McGuire et al., 2003*: flies were kept for 5 days at 29 °C before staining, conditioning and until memory test for LTM analysis for RNAi-Spn expressing flies and 9 days for flies expressing RNAi-Arl8.

## Immunostaining

Larval dissections and immunostaining were performed as previously described (*Qin et al., 2005*). Homeostatic plasticity was induced through pharmacological challenge with 50 µM PhTx (Philantho-toxin 433 TFA salt, Aobious, CAS no. 276684-27-6) in calcium-free HL3 at room temperature. Controls were similarly treated by substituting PhTx with dH20. Briefly, the larvae were immobilized with insect pins on a rubber dissection pad, cut open dorsally between the dorsal tracheal trunks, avoiding excessive stretching or tissue damage. The semi-intact larvae were incubated with PhTx for 10 or 30 min. The preparation was completed by flattening the body wall using insect pins to expose the muscles. In case of Lat-B, the semi-intact preparations were treated for 10 min; and in case of Rok inhibitor, larvae were treated for 10 min followed by treatment with PhTx +Rok solution.

For staining against Unc13A (*Böhme et al., 2016*) larvae were fixed in methanol for 5 min. For BRP (Nc82, Developmental Studies Hybridoma Bank, RRID: AB_2314866), Spn (*Muhammad et al., 2015*), Mical (*Grintsevich et al., 2016*) and Syt-1 (Developmental Studies Hybridoma Bank, RRID: AB_528483) immunostaining, larval filets were fixed with 4%- paraformaldehyde (PFA) in 0.1 mM phosphate-buffered saline (PBS) for 10 min. Secondaries used were Goat anti-mouse Alexa Fluor-488 (Invitrogen, Cat#A-11001; RRID: AB_2534069), Goat anti-rabbit-Cy3 (Jackson ImmunoResearch, Cat#111-165-144; RRID: AB_2338006) and Anti-Horseradish Peroxidase Alexa Fluor-647 (Jackson ImmunoResearch, Cat#123-605-021; RRID: AB_2338967). Larvae were then processed for immuno-histochemistry and mounted in Vectashield (Vector Labs, CA, USA). For STED imaging, larvae were mounted in ProLong Gold Antifade Mountant (Thermo Fisher Scientific).

Brains were dissected in ice-cold hemolymph-like saline (HL3; composition in mM: NaCl 70, KCl 5, MgCl2 20, NaHCO3 10, trehalose 5, sucrose 115, HEPES 5, pH adjusted to 7.2) solution and immediately fixed in 4% paraformaldehyde (PFA) for 30 min at room temperature under stirring. Samples were incubated for 2 hr in phosphate-buffered saline (PBS) containing 1% Triton X-100 (PBT) containing 10% normal goat serum (NGS). Subsequently, the samples were incubated in the primary antibody solution diluted in PBT-5 % NGS at 4 °C under stirring for 48 hr. Samples were washed six times for at least 30 min each in PBT at room temperature, and subsequently incubated with secondary antibody solution diluted in PBT-5 % NGS at 4 °C overnight. Brains were washed at room temperature six times for at least 30 min each in PBT. Finally, the mounting was done in Vectashield (Vector Laboratories) on glass slides.

## Image acquisition, processing, and analysis

Confocal microscopy was performed with a Leica TCS SP8 inverted confocal microscope (Leica DMI 6000, Leica Microsystems, Germany) and STED microscopy with a Leica TCS SP5 microscope. All images were acquired at room temperature using LCS AF software. (Leica Microsystems, Germany). Confocal imaging was performed using a 63×1.4 NA oil immersion objective. Images of muscle 4 Type-1b NMJs were obtained from abdominal segments A3-A4 of the fixed larval preparations for all experiments. Images were acquired in line scanning mode with a pixel size 75.16 nm*75.16 nm and with a z step of 0.25 µm. Stacks were processed with Fiji (*Schindelin et al., 2012*) software. Images were quantified for average intensity of over an NMJ. In vivo live imaging was performed using a Leica SP8 microscope and a 63×1.4 NA oil immersion objective. Images of muscle 26 and 27 Type-1b NMJs were obtained from larval abdominal segments A2-A4. Confocal images were acquired in line scanning mode with a pixel size of 75.16 nm*75.16 nm and with a z step of 0.25 µm. Images were obtained from third instar larvae by immobilizing them in an airtight imaging chamber with a small amount of Voltalef H10S oil (Arkema, Inc, France). The larvae were anaesthetized with short pulses of a mixture of air and desflurane (Baxter,IL, UAS).

Adult brain samples were imaged using Leica SP8 confocal microscope equipped with x20 apochromat oil-immersion Leica objective (NA = 0.75) and x40 apochromat oil-immersion objective (NA = 1.30). Alexa Fluor 488 was excited at 488 nm, Cy3 at 561 nm and Cy5 at 633 nm wavelengths.

Samples were scanned using LAS X software (3.5.2.18963) at 0.5 μm sections in the z direction. All images were acquired at 8-bits grayscale. Segmentation of the image stacks were processed using the Amira software (Visage Imaging GmbH). The first step was to define a unique label for each region in the first (BRP$^{nc82}$) fluorescent channel for the RNAi experiments and the second (Syd-1) fluorescent channel for the experiments after conditioning. A full statistical analysis of the image data associated with the segmented materials was obtained by applying the Material Statistics module of the Amira software, in which the mean gray value of the interior region is calculated. To avoid difference of global staining between different groups, the intensity of the staining of the MB lobes and calyx was normalized on the intensity of Antennal Lobes (AL) and Protocerebral Bridge (PB) respectively for quantification purpose. The median voxel values of the regions were compared, as measured in individual adult brains, in order to evaluate the synaptic marker label.

All brain representative images were processed using the ImageJ/Fiji software (1.52 P, https://imagej.net/software/fiji/) for adjusting brightness with the brightness/contrast function. Images shown in a comparative figure were processed with exactly the same parameters.

## Electrophysiology

Two-electrode voltage clamp (TEVC) as well as single electrode (current-clamp) recordings were performed at room temperature on muscle 6 of 3rd instar larval NMJs in the abdominal segments A2 and A3. Larvae were reared on normal food or food containing cycloheximide as indicated. Third instar larvae were dissected in modified $Ca^{2+}$-free hemolymph-like saline (HL3; in mM: NaCl 70, KCl 5, $NaHCO_3$ 10, $MgCl_2$ 20 (TEVC) or 10 (current clamp), Sucrose 115, Trehalose 5, HEPES 5). For PhTx experiments, larvae were cut open along the midline and incubated for the indicated time (10 or 30 min) in HL3 containing either 50 μM PhTx or the equivalent volume of water for controls. For experiments using latrunculin A, the solution also contained the indicated amount of latrunculin A (or water for controls); additionally, open preparations were pre-incubated for 10 min in HL3 containing only latrunculin A in the indicated concentration (or water for controls) and no Phtx. The solution used for this pre-incubation was then removed and HL3 containing both latrunculin A and Phtx (or equal volumes of water for controls) added. The incubation solution was gently perfused into the preparation using a pipette at the start of the incubation and once more at the halftime point (after 5 or 15 min). The preparation was then finished during the last 2 min of incubation time and afterwards washed 3 times with HL3 before being transferred to bath solution for electrophysiological recordings. Recordings were obtained with a bath solution of HL3 with 1.5 (TEVC) or 0.4 (current clamp) mM $CaCl_2$. Recordings were made from cells with an initial Vm between –50 (TEVC) or –40 (current clamp) and –80 mV, and input resistances of ≥4 MΩ, using intracellular electrodes with resistances of 30–50 MΩ, filled with 3 M KCl. 2 Cells were recorded per animal. Glass electrodes were pulled using a Flaming Brown Model P-97 micropipette puller (Sutter Instrument, CA, USA). Recordings were made using an Axoclamp 2 B amplifier with HS- 2Ax0.1 head stage (Molecular Devices, CA, USA) on a BX51WI Olympus microscope with a 40 X LUMPlanFL/IR water immersion objective (Olympus Corporation, Shinjuku, Tokyo, Japan). mEJCs/mEPSPs were recorded for 90 seconds with the voltage clamped at –80 mV (for TEVC recordings), all other recordings were performed while clamping the voltage at –60 mV (for TEVC recordings). eEJCs/eEPSPs were recorded after stimulating the appropriate motor neuron bundle with 5 (eEJCs) or 8 (eEPSPs) V, 300 μs at 0.2 Hz using an S48 Stimulator (Grass Instruments, Astro-Med, Inc, RI, USA). To estimate the RRP size, single train recordings with 61 stimulations were performed at 100 Hz. Signals were digitized at 10 kHz using an Axon Digidata 1322 A digitizer (Molecular Devices, CA, USA) and low pass filtered at 1 kHz using an LPBF-48DG output filter (NPI Electronic, Tamm, Germany). The recordings were analyzed with pClamp 10 (Molecular Devices, Sunnyvale, CA, USA), GraphPad Prism 6 (GraphPad Software, Inc, San Diego, CA, USA) and two Python scripts utilizing the pyABF package for Python 3.10 (Harden, SW (2022). pyABF 2.3.5. [Online]. Available: https://pypi.org/project/pyabf). Stimulation artifacts of eEJCs/eEPSPs were removed for clarity. mEJCs/mEPSPs were further filtered with a 500 Hz Gaussian low-pass filter. Using a single template for all cells, mEJCs/mEPSPs were identified and averaged, generating a mean mEJC/mEPSP trace for each cell. An average trace was generated from 20 eEJC/eEPSP traces per cell for 0.2 Hz stimulation and 10ms ISI paired pulse recordings and from 10 traces for 30ms ISI paired pulse recordings. Rise time was calculated from the average trace of the 0.2 Hz stimulation recording as the time from 10% to 90% of the total amplitude before the peak. Decay constant $\tau$ was calculated by fitting

a first order decay function to the region of the average trace of the 0.2 Hz stimulation recording from 60% to 5% of the total amplitude after the peak. The amplitude of the average eEJC/eEPSP trace from the 0.2 Hz stimulation recording was divided by the amplitude of the averaged mEJC/mEPSP, for each respective cell, to determine the quantal content. 10ms and 30ms ISI paired pulse ratios were calculated by dividing the amplitude after the second pulse by the amplitude after the first pulse. The baseline for the second amplitude was set at the last point before the second stimulation artifact onset. Estimation of the RRP size and refilling rates were performed as described previously (Matkovic et al., J Cell Biol. 2013 Aug 19; 202(4): 667–683.); briefly, the amplitudes of the responses to each stimulation in the train were extracted from the recording by using the last data points before each artifact onset as a baseline. For each cell, these amplitudes were then added cumulatively and divided by the average mEJC amplitudes as measured for this cell to obtain the cumulative quantal content. This cumulative quantal content was then plotted against the number of stimulations and a linear regression was performed for the last 20 amplitudes, where the cells reach a steady state due to the depletion of the RRP. The intersect of this linear fit with the Y-axis is the estimated RRP size, the slope of the linear fit is the estimated refilling rate.

## Behavior: olfactory associative aversive conditioning

Flies were trained using the classical olfactory aversive conditioning protocols described by *Tully and Quinn, 1985*. Training and testing were performed in climate-controlled boxes at 25 °C in 80% humidity under dim red light. At 2–3 days old, flies were transferred to fresh food vials and either put at 29 °C for RNAi induction or stayed at 18 °C for the non-induced controls. Conditioning was performed on groups of around 40–50 flies with 3-octanol (around 95% purity; Sigma-Aldrich) and 4-methylcyclohexanol (99% purity; Sigma-Aldrich). Odors were diluted at 1:100 in paraffin oil and presented in 14 mm cups. A current of 120 AC was used as a behavioral reinforcer. Memory conditioning and tests were performed with a T-maze apparatus (*Tully and Quinn, 1985*). In a single-cycle training, groups of flies were presented with one odor (CS$^+$) paired with electrical shock (US; 12 times for one minute). After one minute of pure air-flow, the second odor (CS$^-$) was presented without the shock for another minute. During the test phase, flies were given 1 min to choose between 2 arms, giving each a distinct odor. An index was calculated as the difference between the numbers of flies in each arm divided by the sum of flies in both arms. The average of two reciprocal experiments gave a performance index (PI). The values of PI ranges from 0 to 1, where 0 means no learning (50:50 distribution of flies) and a value of 1 means complete learning (all flies avoided the conditioned odor). For olfactory acuity and shock reactivity, around 50 flies were put in a choice position between either one odor and air for one minute or electric shocks and no-shocks, respectively.

## Electron microscopy: conventional embedding of larval muscles

Third instar larvae were dissected as described above. Control animals were incubated in HL3 solution with or without PhTx for 3 min and *Spn* mutants were incubated with or without PhTx for 5 min. The larvae were fixed with 4% PFA and 0.5% glutaraldehyde (GA) for 10 min. The larval filets were collected on ice in 2% GA in 0.1 M sodium cacodylate buffer (NaCac). The filets were then fixed for 1 hr in 2% GA in 0.1 M NaCac at RT. The samples were washed thrice with NaCac buffer, allowing the filets to rinse for 5 min each time. The filets were transferred to Snap-On lid vials with the last washing step and were kept in the dark, in ice and on a shaker after this point. The filets were post-fixed with 1% OsO$_4$ in 0.8% KFeCn for 1 hr. The samples were washed for 1 hr with NaCac buffer followed by three washes with Millipore water. The samples were incubated for 1 hr in 1% uranyl acetate (UrAc). The samples were then dehydrated through a series of increasing alcohol concentrations and embedded in EPON resin by incubating in 1:1 EtOH/EPON solution and then in pure EPON. Quantification of EM data was done by demarcating an area around a T-bar which remained the same across genotypes and counting the number of vesicles.

## Quantification and statistical analysis

Data was analysed using Prism (GraphPad Software, CA, USA). The experiments were performed in duplicate and when necessary, in triplicate biological replicates. Outliers were excluded using the ROUT method. In all data sets with two groups, unpaired t-test was performed. Unpaired t-test with Welch's correction which was used when the standard deviations could not be assumed to be

equal amongst the datasets, as determined by F-test for comparing the variances or standard deviations from two populations. For all data sets with three or more groups, one-way analysis of variance (ANOVA) followed by Tukey's multiple comparison test was used. As a test for normality, D'Agostino & Pearson test was utilized. For all data failing normality test, Mann-Whitney U test was used and two-way ANOVA followed by Holm-Sidak's multiple comparisons test was used for grouped analyses. Statistical parameters are stated in the figure legends and all values of mean, SEM and statistical parameters are provided in the source data tables. Data is represented as mean ± SEM. Statistical significance is denoted in the graphs as asterisks: *, $p < 0.05$; **, $p < 0.01$; ***, $p < 0.001$; ns. (not significant), $p > 0.05$.

For adult brain staining experiments, differences among multiple groups were tested by one-way ANOVA with Tukey's post hoc test, whereas differences between two groups were test by t-test. Memory scores are displayed as mean ± SEM. For behavioral experiments, scores resulting from all genotypes were analyzed using one-way ANOVA followed, if significant at $p \leq 0.05$, by Tukey's multiple-comparisons tests. For memory experiments, the overall ANOVA p-value is given in the legends, along with the value of the corresponding Fisher distribution F(x,y), where x is the number of degrees of freedom for groups and y is the total number of degrees of freedom for the distribution. Asterisks on the figure denote the least significant of the pairwise post hoc comparisons between the genotype of interest and its controls following the usual nomenclature (ns, (not significant) $p > 0.05$; *, $p \leq 0.05$; **, $p \leq 0.01$; ***, $p \leq 0.001$).

## Proteomics

Co-IP of Bruchpilot and Spinophilin Pulldown experiments were performed with crude synaptosomes resuspended in homogenization buffer (320 mM Sucrose, 4 mM HEPES and a protease inhibitor cocktail, pH 7.2). Approximately 6000 fly heads were collected per replicate, and synaptosomes were purified via differential centrifugation (see *Depner et al., 2014*). Twenty µg of each antibody (rbBRPlast200 and rbSPN) was coupled to 50 µl Protein A–coated agarose beads. For a negative control rb-IgGs were coupled to beads. To avoid unspecific bounds of proteins to beads, synaptosome suspension was precleared by rotating for 1 h at 4 °C on naked beads. Afterwards, bead-antibody and bead-IgG complexes were incubated with solubilized and precleared synaptosome membrane preparations (P2) overnight at 4 °C. After four washing steps with IP buffer (containing 20 mM Hepes, pH 7.4, 200 mM NaCl, 2 mM MgCl2, 1 mM EGTA and 1% Triton X-100), antibody antigen complexes were eluted with 60 µl 2×denaturing protein sample buffer each.

The data output from Maxquant was analysed using Perseus software (version 1.6.2.3). LFQ values were log2 transformed to achieve normal data distribution. Proteins identified in at least three (out of four) replicates were considered for statistical analysis all other proteins that were detected and quantified in only one replicate were excluded. Missing data were imputed by values from a normal distribution (width 0.3 standard deviations; down shift 1.8). For statistical protein enrichment analysis in the BRP-IP or SPN-IP, a two-sided t-test between BRP-IP or SPN-IP and negative IgG control was used with a S0 constant of 0.1 and a permutation-based FDR of 0.05. Presented fold changes have been calculated as difference from mean values of log2 transformed intensities from BRP-IP or SPN-IP and IgG control. Microsoft Excel was used to create Volcano plot from quadruplicates of coprecipitated protein levels from the BRP-IP or SPN-IP compared with the IgG control. The x-axis represents the log2 fold-change value, indicating the magnitude of change, and the y axis is –log10 of the p-value showing statistical significance.

Pathway and gene ontology (GO) enrichment analysis were carried out using Metascape 3.5 tool (http://metascape.org). Here, enriched GO terms were shown in a heatmap using a color scale to represent statistical significance (darkness reflects the p-value of the given GO term).

## Acknowledgements

SJS gratefully acknowledges support by grants from the Deutsche Forschungsgemeinschaft [EXC 2049 (#390688087); NanoSYNDIV (SI 849/10–1); TRR186/A03 (#278001972), NeuroNex2 (#436260754), and FOR5228 (#447288260) to SJS].

## Additional information

### Funding

| Funder | Grant reference number | Author |
|---|---|---|
| Deutsche Forschungsgemeinschaft | EXC 2049 (#390688087) | Stephan J Sigrist |
| Deutsche Forschungsgemeinschaft | TRR186/ A03 (#278001972) | Stephan J Sigrist |
| Deutsche Forschungsgemeinschaft | NeuroNex2 (#436260754) | Stephan J Sigrist |
| Deutsche Forschungsgemeinschaft | FOR5228 (#447288260) | Stephan J Sigrist |
| European Research Council | [ERC-AdG "Synprotect"] | Stephan J Sigrist |
| NanoSYNDIV | SI 849/10-1 | Oriane Turrel |

The funders had no role in study design, data collection and interpretation, or the decision to submit the work for publication.

### Author contributions

Niraja Ramesh, Conceptualization, Data curation, Formal analysis, Validation, Investigation, Visualization, Methodology, Writing – original draft, Project administration, Writing – review and editing; Marc Escher, Oriane Turrel, Janine Lützkendorf, Data curation, Formal analysis, Investigation; Tanja Matkovic, Data curation, Investigation; Fan Liu, Supervision; Stephan J Sigrist, Conceptualization, Resources, Data curation, Supervision, Funding acquisition, Visualization, Methodology, Writing – original draft, Project administration, Writing – review and editing

### Author ORCIDs

Niraja Ramesh ⬡ http://orcid.org/0000-0002-7099-4093
Oriane Turrel ⬡ http://orcid.org/0000-0003-2174-7659
Janine Lützkendorf ⬡ http://orcid.org/0000-0001-5051-6281
Stephan J Sigrist ⬡ http://orcid.org/0000-0002-1698-5815

### Decision letter and Author response

Decision letter https://doi.org/10.7554/eLife.86084.sa1
Author response https://doi.org/10.7554/eLife.86084.sa2

## Additional files

### Supplementary files

• MDAR checklist

### Data availability

All data generated or analysed during this study are included in the manuscript and supporting file; Source Data files have been provided for all figures.

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
