## [Editor Report]

The study advances mechanistic understanding of presynaptic plasticity (PHP): a process through which presynaptic nerve terminals adjust the strength of their output and which underpins key aspects of memory and learning. The authors use the well-characterised and tractable *Drosophila* NMJ to identify two key proteins – Spinophilin (Spn) and Syd-1 (a Rho GTPase activating protein) and show that the antagonism that occurs between these two components is sufficient to regulate F-actin stability at the synapse. Destabilization of F-actin, required for the maintenance of PHP, promotes synaptic vesicle release.

---

## [Decision Letter]

**Decision letter after peer review:**

Thank you for submitting your article "An antagonism between Spinophilin and Syd-1 operates upstream of memory-promoting presynaptic long-term plasticity" for consideration by *eLife*. Your article has been reviewed by 2 peer reviewers, and the evaluation has been overseen by Mani Ramaswami as Reviewing Editor and Gary Westbrook as the Senior Editor. The following individual involved in the review of your submission has agreed to reveal their identity: Richard A Baines (Reviewer #1).

Essential revisions:

The reviewers were impressed by the work, as well as the quality and interest of your observations and conclusions. However, there was a concern that the conclusion that Spn regulates synaptic plasticity through Mical-dependent control of actin dynamics was insufficiently supported, and also that the connection between NMJ mechanisms and MB mechanisms needs further clarification. The main revisions requested are listed below, but please also provide itemised responses to the other listed reviewer concerns (e.g, the one pertaining to mass spec data).

1) Spn, Syd1 double mutants can support PHP: if this suppression is through Mical, then one would predict that Mical levels should also be normalized in the Spn, Syd1 double mutants. Please include data that addresses this issue. If Mical levels are not normalised then the conclusions should be appropriately moderated to simply state that "Spn regulates synaptic plasticity through Syd-antagonism and through other Syd-independent pathways."

2) During olfactory conditioning: (a) synaptic changes are restricted to specific compartments of the MB; and (b) unlike homeostatic potentiation of transmitter release at the NMJ, one observes depression of transmitter release from Kenyon cell terminals.

This makes it necessary to: (i) examine and report whether Brp and Unc13A levels are specific to compartments in the MB that participate in aversive conditioning (if this has been previously done, then please refer to the appropriate publication and relevant figure); and (ii) explain how Syd-1 could be involved in both presynaptic facilitation (NMJ) and depression (MB).

*Reviewer #1 (Recommendations for the authors):*

In general, the work appears to be of high quality and the conclusions drawn fit the data shown. However, this is not an easy paper to read and requires the reader to continually move between two sets of figures and a data table. Case in point is Figure 5f, expression of UAS-Mica(δ) redox: the effect on QC is reported as ns, but there is clearly an increase in QC in the presence of this transgene that looks 'similar to' the increase reported as significant for UAS-Mical. P values in the data table indicate P = 0.0056, 0.0224, and 0.2519, respectively (thus supporting the stated conclusion). The latter is tested using a t-test (normalised data), whilst the former two by Mann-Whitney (non-normal). Given this is the same type of data, why are some normally distributed when others are not? Moreover, the statistics section in the methods reports a comparison of 3 or more groupings that were analysed by ANOVA. The graph as presented suggests all data was analysed at the same time, but one must assume that they were analysed as 3 x pairings. Perhaps breaks in the X-axis could be used to denote separate data. It would also aid the reader if figure legends for main text figures could be enhanced to include actual P values, n, and stats tests used, at least for certain data (as described above). I appreciate, however, that we all must adhere to journal-style requirements.

Indeed, the data table sometimes lists a t-test alone as being used, but sometimes with Welch's correction utilised for what seems to be identical data (Figure 5D). The latter correction is not described in the methods section. The inclusion of a description of why data from the same synapse might differ in its distribution (normal vs not-normal) would be helpful for those not expert in this preparation.

Specific points:

With reference to the increase in QC in spn (10 min PHP, Figure 1i) – the variability of the observed increase is greater than that of the *w1118* control (being sig diff at only 0.05). Whilst this does not detract from the overall conclusion that spn is not required for PHP induction, the greater variance suggests that it does, nevertheless, have some organising role at this earlier time point. This point should be better discussed as the conclusion of this experiment, as presented, is too binary (no effect on PHP induction, required for maintenance). Note the use of black circles for w118 in figures often obscures the average/SEM bars. Perhaps grey could be used for the circles and black for the bars?

Whilst presynaptic expression of Spn is sufficient to rescue the Spn phenotype at the NMJ, did the authors also test post-synaptic rescue? Could it be the case that Spn can be localised either pre or post to have an effect?

Cycloheximide is reported to have no effect on the activity of Spn. However, without a positive control, it is difficult to know if this experiment was conducted appropriately – How to be sure that the feeding regime / acutely used dose was adequate?

It would be interesting to speculate whether Spn directly binds Syd-1 or not. Syd-1 is shown in Figure 3B – but not discussed in the results – is this a significant level of Syd-1? The legend of Figure 3 refers the reader to Figure s3 – but there is no proteomic data displayed in the latter.

---

## [Author Response]

Essential revisions:1) Spn, Syd1 double mutants can support PHP: if this suppression is through Mical, then one would predict that Mical levels should also be normalized in the Spn, Syd1 double mutants. Please include data that addresses this issue. If Mical levels are not normalised then the conclusions should be appropriately moderated to simply state that " Spn regulates synaptic plasticity through Syd-antagonism and through other Syd-independent pathways."

We thank the reviewer for this comment and agree that, according to our data and conclusions, we would predict that Mical levels should be normalized upon *syd-1* heterozygosity in S*pn* mutants. To test this, we now immunostained for Mical in wildtype, *Spn* mutants and *Spn* mutants with *syd-1* heterozygosity. Indeed, in *Spn* mutants with *syd-1* heterozygosity, Mical levels were comparable to wildtype NMJs (Figure 5 —figure supplement 1 c-d). We have added this novel finding to the manuscript as follows:

“Interestingly, NMJ Mical levels are also subject to Spn/Syd-1 antagonism. *Syd-1* heterozygosity in *Spn* mutants resulted in a reestablishment of Mical levels comparable to wildtype NMJs (Figure 5 —figure supplement 1 c-d).”

2) During olfactory conditioning: (a) synaptic changes are restricted to specific compartments of the MB; and (b) unlike homeostatic potentiation of transmitter release at the NMJ, one observes depression of transmitter release from Kenyon cell terminals. This makes it necessary to: (i) examine and report whether Brp and Unc13A levels are specific to compartments in the MB that participate in aversive conditioning (if this has been previously done, then please refer to the appropriate publication and relevant figure)

We appreciate the opportunity to discuss this fully justified comment in detail, and are aware that our findings open up questions that need to be worked out in the future. As correctly pointed out, the *acute, immediate* formation of aversive short-term memory (STM) indeed was previously shown to trigger synaptic depression at the KC>MBON synapse in the respective MB compartment (Cohn, Morantte, and Ruta 2015; Hige et al. 2015; Owald and Waddell 2015). Of note here, our genetic manipulations do not affect the *acute, immediate* formation of aversive STM, therefore we do not a priori assume that the Spn-mediated AZ remodeling results in synaptic depression. We favor the idea that our plasticity mechanism executes a post-conditioning aftermath critical for consolidating memories over time. This derives from the consideration that synaptic weights, in all likelihood, have to get re-normalized in the MB circuitry after the acute, immediate STM encoding depression. To reiterate, AZ remodeling is indeed fully dispensable for STM, as shown in this manuscript, as well as in Turrel et al. (Turrel et al. 2022). Instead, we find a critical role for AZ remodeling for longer term (1 hr and 3 hr olfactory memory), in both this manuscript (Figure 7 a-d, S8 a-b) and also in our previous analysis (Turrel et al. 2022). We thus think that one possibility to explain these findings is that the BRP/Unc13A increases that we observe promote potentiation in the MB network to facilitate such a renormalization. In order for individual KCs to reestablish their overall excitation/inhibition balance, such regulation would not necessarily have to be compartment-specific.

Still, we fully admit that the exact relation of the AZ remodeling described in this paper, to STM depression specifically, and KC presynaptic plasticity in general, is currently unknown. Specifically, we cannot at this time tell (i) whether the conditioning-associated presynaptic remodeling is indeed potentiating KC and MB AZs, (ii) whether at all overlapping sets of synapses are involved in STM and MTM formation and (iii) consequently whether our longer-term AZ remodeling should be compartment specific. Protocols which would allow us to satisfactorily follow synapse physiology along the memory forming and conditioning time axis by using electrophysiology, are currently non-existing and that their development would be challenging. What we *can* say, however, is that the molecular machinery which executes structural remodeling at NMJ AZs (here associated with a homeostatic stabilization of NMJ transmission strength due to an increase of quantal content) is critically needed for MTM formation within the MB intrinsic neurons.

We did observe specificity of our AZ plasticity mechanism in the distinct MB lobe systems, in our previous work. Notably, the distinct MB lobe systems have been found to have different importance for the distinct memory components. STM and LTM were shown to be encoded in γ KCs and α/β KCs respectively (Pascual and Preat 2001; Qin et al. 2012; Blum et al. 2009). Previous reports have shown that early memory requires transmission from the α/β, α′/β′, and γ KCs (Cervantes-Sandoval et al. 2013; Wang et al. 2008; McGuire, Le, and Davis 2001). Both α/β and γ KCs are also involved in memory retrieval at 2–3 hr (Xie et al. 2013; Isabel, Pascual, and Preat 2004) when both ARM and MTM can be measured. Concerning the specificity of AZ remodeling, distinct MB lobes are of importance for the role of AZ remodeling in MTM, as in Turrel et al. we showed that α/β lobe neurons were the most important as described in the following paragraph (Turrel et al. 2022) (Figure 1 A-G):

“1 h after training, animals displayed BRP and even more pronounced Unc13A level increases. At 3 h post training, average MB BRP levels plateaued, whereas Unc13A levels increased further (Figure 1B). Syd-1 levels reached a significant increase only 3 h after training (Figure 1B). We indeed found that the kinetics and extent of AZ changes differed here: in α/β lobes, BRP and Unc13A levels increased 1 and 3 h after conditioning, whereas Syd-1 levels increased only at 1 h (Figure 1D). In α’/β’ neurons, BRP, Syd-1, and Unc13A increased at 1 and 3 h (Figure 1E). In γ lobes, Unc13A only increased at 3 h, whereas BRP showed an increase at 1 h but went back to baseline at 3 h (Figure 1F), the latter mirroring the changes observed using the STAR system by Zhang et al. (Zhang et al. 2018). Notably, at the MB calyx, no significant intensity changes were observed after paired conditioning (Figure 1G), suggesting that the changes are at least particularly pronounced in the MB lobes dominated by the AZ of KCs. However, MB lobes staining also contains contributions of other neuronal populations, particularly dopaminergic neurons (DANs). To test for the specific contribution of the MB KCs, we expressed an RNA interference (RNAi) construct against BRP using the KC generic driver OK107-Gal4. Consequently, different from *w1118* and Gal80ts;OK107/+ controls, neither BRP nor Unc13A levels increased significantly after paired conditioning in flies expressing a BRP-directed RNAi in adult MB KCs, suggesting an important role for BRP driven plasticity at KC AZs for this conditioning-associated plasticity (Figure S1C).”

Our strategy for this work was to use the NMJ PhTx assay to identify proteins involved in AZ remodeling that could also be involved in memory formation in adult flies. As said, as of now, we have no experimental evidence of whether the AZ remodeling observed in the MB actually leads to synaptic depression, or whether it is instead downstream of the initial short-term synaptic depression. This study and Turrel et al. (Turrel et al. 2022), however, provide evidence for an overlap of the executory machinery involved in both NMJ PHP plasticity and MTM formation, as BRP, Spn, Arl8, IMAC and Aplip1 are involved specifically both in mid-term NMJ PHP (at 30 min after PhTx treatment) and in MTM.

ii) Explain how Syd-1 could be involved in both presynaptic facilitation (NMJ) and depression (MB).

This is an important and particularly interesting aspect of this work. We previously found that Syd-1 levels are significantly increased 1 h after conditioning in in the α/β and α’/β’ lobes (Turrel et al. 2022), while Syd-1 levels are not increased 10 min after PhTx treatment at the NMJ (Bohme et al. 2019). This finding indicates that some of the AZ proteins may be affected differently in the two plasticity processes i.e. they are not always modulated identically. Most importantly, however, we consistently find that Syd-1 counteracts Spn-executed structural, functional and behavioral plasticity: (i) loss of a single *Syd-1* gene copy “rescues” the PhTx triggered PHP and BRP remodeling deficit of *Spn* mutants. Here, *Syd-1* mutants continue to show elevated levels of BRP whereas wildtype animals show a transient increase in BRP followed by a return to baseline (Figure 2 m-n). (ii) reduction of Syd-1 rescued the memory display/formation provoked by Spn knockdown (Figure 7k). Lacking a temporally extended in vivo electrophysiological analysis on memory associated changes, we cannot determine the exact functional changes associated with memory consolidation, the process we show to depend on AZ remodeling controlled by the Spn/Syd-1 antagonism.

Reviewer #1 (Recommendations for the authors):In general, the work appears to be of high quality and the conclusions drawn fit the data shown. However, this is not an easy paper to read and requires the reader to continually move between two sets of figures and a data table. Case in point is Figure 5f, expression of UAS-Mica(δ) redox: the effect on QC is reported as ns, but there is clearly an increase in QC in the presence of this transgene that looks 'similar to' the increase reported as significant for UAS-Mical. P values in the data table indicate P = 0.0056, 0.0224, and 0.2519, respectively (thus supporting the stated conclusion). The latter is tested using a t-test (normalised data), whilst the former two by Mann-Whitney (non-normal). Given this is the same type of data, why are some normally distributed when others are not? Moreover, the statistics section in the methods reports a comparison of 3 or more groupings that were analysed by ANOVA. The graph as presented suggests all data was analysed at the same time, but one must assume that they were analysed as 3 x pairings. Perhaps breaks in the X-axis could be used to denote separate data. It would also aid the reader if figure legends for main text figures could be enhanced to include actual P values, n, and stats tests used, at least for certain data (as described above). I appreciate, however, that we all must adhere to journal-style requirements.Indeed, the data table sometimes lists a t-test alone as being used, but sometimes with Welch's correction utilised for what seems to be identical data (Figure 5D). The latter correction is not described in the methods section. The inclusion of a description of why data from the same synapse might differ in its distribution (normal vs not-normal) would be helpful for those not expert in this preparation.

We would like to sincerely thank the reviewer for the in depth analysis of our data. The reviewer is right in observing that there is a statistically insignificant increase in QC upon UAS-MicaldRedox expression in *Spn* mutants, while there are statistically significant increases in QC upon UAS-Mical or UAS-Spn expression in *Spn* mutants when 3 separate t tests were performed (Figure 5). When ANOVA was performed on the raw data comparing all 6 groups simultaneously, there were no statistically significant changes between any of the groups (Figure S6). Importantly, however, the recovery in evoked release upon UAS-Mical and UAS-Spn expression in *Spn* mutants was very clear in both the normalized and the raw data. In contrast, there was no recovery of evoked release upon UAS-MicaldRedox expression in *Spn* mutants, supporting our conclusion that the redox activity of Mical is important for PHP. The statistically insignificant increase in QC upon UAS-MicaldRedox expression in *Spn* mutants could result from the fact that QC is calculated from the experimentally measured evoked and mini amplitudes and the mini amplitudes of this genotype are slightly (importantly, statistically insignificantly) lower than PhTx treated UAS-Mical and UAS-Spn groups.

We admit that it is not particularly easy to move between the main and supplementary figures and the text. Most papers addressing PHP at the *Drosophila* NMJ show normalized data in the main figures and raw data as supplementary figures, however. Normalization of the PhTx treated genotypes with the corresponding untreated control aids in assessing the effect on PHP, which is the primary question addressed in this study. Though we already went through efforts to improve the figures as suggested, we are certainly open to suggestions to improving the layout of the figures and thus allow for an easier access to the manuscript.

Furthermore, we have now added further description of the statistical tests used, specifically the usage of unpaired t-test with Welch's correction which was used when the standard deviations cannot be assumed to be equal amongst the datasets, as determined by F-test for comparing the variances or standard deviations from two populations. Furthermore, as a test for normality, D'Agostino and Pearson test was utilized, and Mann-Whitney U test, or 2 way ANOVA followed by Holm-Sidak's multiple comparisons test were performed. The differences in the distribution of the data in terms of normality is likely the result of the relatively small n in these experiments, and addition of more data points would likely have resulted in normal distribution of datasets that are currently statistically tested as non-normal. These tests for normality, variance and statistical significance with corrections are used widely in similar studies.

We have now described the statistical methods used in the manuscript as follows:

“Outliers were excluded using the ROUT method. In all data sets with two groups, unpaired t-test was performed. Unpaired t-test with Welch's correction which was used when the standard deviations could not be assumed to be equal amongst the datasets, as determined by F-test for comparing the variances or standard deviations from two populations. For all data sets with three or more groups, one-way analysis of variance (ANOVA) followed by Tukey’s multiple comparison test was used. As a test for normality, D'Agostino and Pearson test was utilized. For all data failing normality test, Mann-Whitney U test was used and 2 way ANOVA followed by Holm-Sidak's multiple comparisons test was used for grouped analyses. Statistical parameters are stated in the figure legends and all values of mean, SEM and statistical parameters are provided in table S1. Data is represented as mean ± SEM. Statistical significance is denoted in the graphs as asterisks: *, p < 0.05; **, p < 0.01; ***, p < 0.001; ns. (not significant), p > 0.05.”

Specific points:With reference to the increase in QC in spn (10 min PHP, Figure 1i) – the variability of the observed increase is greater than that of the w1118 control (being sig diff at only 0.05). Whilst this does not detract from the overall conclusion that spn is not required for PHP induction, the greater variance suggests that it does, nevertheless, have some organising role at this earlier time point. This point should be better discussed as the conclusion of this experiment, as presented, is too binary (no effect on PHP induction, required for maintenance).

The reviewer is fully right in suggesting that Spn has a general role to play in AZ organization and therefore may have an organizing, albeit not strictly essential, role for PHP upon 10min PhTx treatment as well. However, the variance visible in the normalized data is not reflected in the raw data in Figure 1 —figure supplement 1 j. We have now tempered our statement about Spn’s role at this earlier time point in the discussion as follows:

“Like BRP itself, Spn is *per se* dispensable for PHP induction in the tested paradigms. This said, however, a relatively bigger variance in the evoked release responses of *Spn* mutants was observed (Figure 1h), indicating that *Spn* mutants are not completely unaffected also concerning the induction phase of homeostatic plasticity. Induction and maintenance mechanisms might indeed be linked in more complex ways, which remain to be worked out in the future.”

Whilst presynaptic expression of Spn is sufficient to rescue the Spn phenotype at the NMJ, did the authors also test post-synaptic rescue? Could it be the case that Spn can be localised either pre or post to have an effect?

The reviewer here suggested an important experiment. We reported previously that presynaptic expression of Spn rescues the developmental *Spn* phenotype at the NMJ (Muhammad et al. 2015), while postsynaptic expression does not (data not shown). The evoked potentials in PhTx-untreated controls remain low upon muscle expression of Spn in *Spn* mutants, i.e. the baseline phenotype of *Spn* mutants was not rescued by postsynaptic muscle expression (previous unpublished data and Figure 1 —figure supplement 2 n). We therefore had not tried muscle expression of Spn in the PhTx experiments in this manuscript, but now did so in the context of the revision. In the new data, *Spn* mutants with *Mef2*-Gal4 driven muscle expression of Spn, when under PhTx challenge, indeed showed a modest increase in quantal content compared to *Spn* mutants. However, the compensation did not occur to the same extent as with motor neuron expression of Spn. Thus, muscle expression of Spn also seems able to somewhat milden the complete loss of Spn for PHP. Furthermore, muscle expression of Spn did not rescue the BRP incorporation deficit of *Spn* mutants during PHP (data not shown). We have now added the following in the manuscript:

“Full-length Spn, exclusively expressed in the postsynaptic muscle cells, using the *Mef2*-Gal4 driver, did not rescue the PHP-dependent BRP incorporation deficit of *Spn* mutants (data not shown). Moreover, in PhTx-untreated controls, *Spn* mutants with muscle expression of full-length Spn had reduced evoked amplitudes when compared to controls (Figure 1 —figure supplement 2 n). Upon PhTx treatment, *Spn* mutants with muscle expression of full-length Spn showed a modest increase in quantal content (Figure 1 —figure supplement 2 c, j-o), however, not to the same extent as with motor neuron expression of Spn. Thus, muscle expression of Spn also seems able to somewhat milden the complete loss of Spn for PHP, albeit through mechanisms independent of BRP. We therefore focused our analysis on presynaptic Spn in the following experiments.”

Cycloheximide is reported to have no effect on the activity of Spn. However, without a positive control, it is difficult to know if this experiment was conducted appropriately – How to be sure that the feeding regime / acutely used dose was adequate?

As this is certainly a relevant concern which we shared, we replicated two different protocols of cycloheximide application used in previous studies: (a) direct application of cycloheximide to the NMJ during a semi-intact preparation as described in Böhme et al., 2019 (Bohme et al. 2019) and (b) feeding of cycloheximide as described in Tully et al., 1994 (Tully et al. 1994). Both protocols allowed for PHP at 10 min in our hands. As the latter protocol in adult flies was shown to eliminate long-term memory (Tully et al. 1994), we feel rather safe with our conclusion. We are afraid that given the absence of in vivo compatible assays for in situ monitoring of translation over a timeframe of only few minutes we cannot do more in the current moment. We now express this limitation with the following sentence in the manuscript:

“Given the lack of assays allowing for acute in vivo monitoring of local neuronal translation in our system, we cannot be fully sure about the effectiveness of our intervention, however.”

It would be interesting to speculate whether Spn directly binds Syd-1 or not. Syd-1 is shown in Figure 3B – but not discussed in the results – is this a significant level of Syd-1?

It is a great suggestion of the reviewer to mention our previous findings at this point, as Spn and Syd-1 do in fact bind each other Figure 5C in (Muhammad et al. 2015). We now mention this explicitly in the discussion:

“PHP in *Spn* mutants could be reinstated by attenuating Syd-1 levels by removing a single *syd-1* gene copy, extending the developmental role of Spn/Syd-1 antagonism into PHP. Interestingly, Spn and Syd-1 also bind each other as shown through yeast-2-hybrid assay (Muhammad et al. 2015) and co-immunoprecipitation (this study). Future work could lead to an understanding on how their physical association steers their complementary functions.”

All highlighted proteins in Figure 3 were selected because they were statistically significantly enriched in the immunoprecipitation.

Blum, A. L., W. Li, M. Cressy, and J. Dubnau. 2009. 'Short- and long-term memory in *Drosophila* require cAMP signaling in distinct neuron types', Curr Biol, 19: 1341-50.

Bohme, M. A., A. W. McCarthy, A. T. Grasskamp, C. B. Beuschel, P. Goel, M. Jusyte, D. Laber, S. Huang, U. Rey, A. G. Petzoldt, M. Lehmann, F. Gottfert, P. Haghighi, S. W. Hell, D. Owald, D. Dickman, S. J. Sigrist, and A. M. Walter. 2019. 'Rapid active zone remodeling consolidates presynaptic potentiation', Nat Commun, 10: 1085.

Cervantes-Sandoval, I., A. Martin-Pena, J. A. Berry, and R. L. Davis. 2013. 'System-like consolidation of olfactory memories in *Drosophila*', J Neurosci, 33: 9846-54.

Chia, P. H., M. R. Patel, and K. Shen. 2012. 'NAB-1 instructs synapse assembly by linking adhesion molecules and F-actin to active zone proteins', Nat Neurosci, 15: 234-42.

Cohn, Raphael, Ianessa Morantte, and Vanessa Ruta. 2015. 'Coordinated and Compartmentalized Neuromodulation Shapes Sensory Processing in *Drosophila*', Cell, 163: 1742-55.

Depner, H., J. Lutzkendorf, H. A. Babkir, S. J. Sigrist, and M. G. Holt. 2014. 'Differential centrifugation-based biochemical fractionation of the *Drosophila* adult CNS', Nat Protoc, 9: 2796-808.

Hige, Toshihide, Yoshinori Aso, Mehrab N Modi, Gerald M Rubin, and Glenn C Turner. 2015. 'Heterosynaptic Plasticity Underlies Aversive Olfactory Learning in *Drosophila*', Neuron, 88: 985-98.

Isabel, G., A. Pascual, and T. Preat. 2004. 'Exclusive consolidated memory phases in *Drosophila*', Science, 304: 1024-7.

Kuzniewska, B., D. Cysewski, M. Wasilewski, P. Sakowska, J. Milek, T. M. Kulinski, M. Winiarski, P. Kozielewicz, E. Knapska, M. Dadlez, A. Chacinska, A. Dziembowski, and M. Dziembowska. 2020. 'Mitochondrial protein biogenesis in the synapse is supported by local translation', EMBO Rep, 21: e48882.

McGuire, S. E., P. T. Le, and R. L. Davis. 2001. 'The role of *Drosophila* mushroom body signaling in olfactory memory', Science, 293: 1330-3.

Muhammad, K., S. Reddy-Alla, J. H. Driller, D. Schreiner, U. Rey, M. A. Bohme, C. Hollmann, N. Ramesh, H. Depner, J. Lutzkendorf, T. Matkovic, T. Gotz, D. D. Bergeron, J. Schmoranzer, F. Goettfert, M. Holt, M. C. Wahl, S. W. Hell, P. Scheiffele, A. M. Walter, B. Loll, and S. J. Sigrist. 2015. 'Presynaptic spinophilin tunes neurexin signalling to control active zone architecture and function', Nat Commun, 6: 8362.

Nakanishi, H., H. Obaishi, A. Satoh, M. Wada, K. Mandai, K. Satoh, H. Nishioka, Y. Matsuura, A. Mizoguchi, and Y. Takai. 1997. 'Neurabin: a novel neural tissue-specific actin filament-binding protein involved in neurite formation', J Cell Biol, 139: 951-61.

Owald, D., and S. Waddell. 2015. 'Olfactory learning skews mushroom body output pathways to steer behavioral choice in *Drosophila*', Curr Opin Neurobiol, 35: 178-84.

Pascual, A., and T. Preat. 2001. 'Localization of long-term memory within the *Drosophila* mushroom body', Science, 294: 1115-7.

Qin, H., M. Cressy, W. Li, J. S. Coravos, S. A. Izzi, and J. Dubnau. 2012. 'Γ neurons mediate dopaminergic input during aversive olfactory memory formation in *Drosophila*', Curr Biol, 22: 608-14.

Ryan, X. P., J. Alldritt, P. Svenningsson, P. B. Allen, G. Y. Wu, A. C. Nairn, and P. Greengard. 2005. 'The Rho-specific GEF Lfc interacts with neurabin and spinophilin to regulate dendritic spine morphology', Neuron, 47: 85-100.

Tully, T., T. Preat, S. C. Boynton, and M. Del Vecchio. 1994. 'Genetic dissection of consolidated memory in *Drosophila*', Cell, 79: 35-47.

Turrel, O., N. Ramesh, M. J. F. Escher, A. Pooryasin, and S. J. Sigrist. 2022. 'Transient active zone remodeling in the *Drosophila* mushroom body supports memory', Curr Biol.

Wang, Y., A. Mamiya, A. S. Chiang, and Y. Zhong. 2008. 'Imaging of an early memory trace in the *Drosophila* mushroom body', J Neurosci, 28: 4368-76.

Weyhersmuller, A., S. Hallermann, N. Wagner, and J. Eilers. 2011. 'Rapid active zone remodeling during synaptic plasticity', J Neurosci, 31: 6041-52.

Xie, Z., C. Huang, B. Ci, L. Wang, and Y. Zhong. 2013. 'Requirement of the combination of mushroom body γ lobe and α/β lobes for the retrieval of both aversive and appetitive early memories in *Drosophila*', Learn Mem, 20: 474-81.

Zhang, X., Q. Li, L. Wang, Z. J. Liu, and Y. Zhong. 2018. 'Active Protection: Learning-Activated Raf/MAPK Activity Protects Labile Memory from Rac1-Independent Forgetting', Neuron, 98: 142-55 e4.